

# Bromine, iodine and sodium in surface snow along the 2013 Talos Dome – GV7 traverse (Northern Victoria Land, East Antarctica)

Niccolò Maffezzoli[1], Andrea Spolaor[2,3], Carlo Barbante[2,3], Michele Bertò[2], Massimo Frezzotti[4], Paul Vallelonga[1]

[1]Centre for Ice and Climate, Niels Bohr Institute, University of Copenhagen, Juliane Maries Vej 30, Copenhagen Ø 2100, Denmark
[2]Ca'Foscari University of Venice, Department of Environmental Science, Informatics and Statistics, Via Torino 155, 30170 Mestre, Venice, Italy
[3]Institute for the Dynamics of Environmental Processes, IDPA-CNR, Via Torino 155, 30170 Mestre, Venice, Italy
[4]ENEA, SP Anguillarese 301, 00123 Rome, Italy

*Correspondence to*: Niccolò Maffezzoli (maffe@nbi.ku.dk)

**Abstract.** Halogen chemistry in the polar regions occurs through the release of sea salt rich aerosols from sea ice surfaces and organic compounds from algae colonies living within the sea ice environment. Measurements of halogen species in polar snow samples are limited to a few sites although they are shown to be closely related to sea ice extent. We examine here total bromine, iodine and sodium concentrations in a series of 2 m cores collected during a traverse from Talos Dome (72°48' S, 159°06' E) to GV7 (70°41' S, 158°51' E), analyzed by Inductively Coupled Plasma Sector Field Mass Spectrometry (ICP-SFMS) at a resolution of 5 cm. We find a distinct seasonality of the bromine enrichment signal in all cores, with maxima during the austral late spring. Iodine showed average concentrations of 0.04 ppb with little variability. No distinct seasonality was found for iodine and sodium.

The transect revealed homogeneous fluxes for the three chemical species along the transect, due to competing effects of air masses originating from the Ross Sea and the Southern Ocean. The flux measurements are consistent with the uniform values of BrO and IO detected from satellite observations. Similar trends are found for annual bromine enrichment and 130-190° E First Year Sea Ice for the 2010-2013 period.

**Keywords**: bromine, iodine, sodium, sea ice, Antarctica, halogens, polar halogen chemistry, Talos Dome.



# 1. Introduction

Halogen elements play an important role in polar boundary layer chemistry. The release of reactive halogen species from sea ice substrates has been demonstrated to be crucial in the destruction of tropospheric ozone layer at polar latitudes (so called Ozone Depletion Events) during springtime (Barrie et al., 1988).

Although the ocean is the major reservoir of bromine, young sea ice surfaces have high salinity and are therefore a source of bromine halides. Brines, salty aqueous solutions and salty blowing snow on sea ice provide highly efficient saline substrates for reactive halogen activation and release in the atmosphere (Saiz-Lopez et al., 2012).

Bromine radicals result from the photolysis of molecular bromine leading to formation of bromine oxide, BrO, through catalytic reactions that destroy ozone molecules. These chemical cycles have been confirmed by satellite Multi AXis-Differential Optical Absorption Spectroscopy (MAX-DOAS) observations of tropospheric BrO over polar regions (Schönhardt et al., 2012). Bromine can then be recycled and re-emitted from halogen-rich condensed phases (sea salt aerosol) or from sea ice surfaces (Pratt et al., 2013), leading to an exponential increase of bromine in the gas phase (Vogt et al., 1996). Such reactions, known as bromine explosions, also lead to enhanced bromine deposition in surface snowpack. Bromine enrichment in snow (compared to sodium, relative to sea water) has therefore been recently used to reconstruct sea ice variability from ice cores both in the Antarctic and Arctic regions (Spolaor et al., 2013b, 2016).

Iodine is emitted by ocean biological colonies and sea ice algae (Vogt et al., 1999; Atkinson et al., 2012) mainly in the form of organic alkyl iodide (R-I) and possibly other compounds. These can be released by wind forced sea spray generation or percolation up to the sea ice surface through brine channels, and are subsequently photolyzed to inorganic species. Plumes of enhanced IO concentrations from satellites and ground based measurements were observed over Antarctic coasts, suggesting a link with biological and chemical sea ice related processes (Schönhardt et al., 2008; Saiz-Lopez et al., 2007). Yet only sporadic events with IO concentrations above detection limits have been observed in the Arctic regions, possibly due to the greater thickness and lower porosity of Arctic sea ice which prevents an efficient release of iodine species in the atmosphere (Mahajan et al., 2010).

Measurements of sea ice related species such as bromine and iodine could therefore allow a sea ice signature to be obtained from ice core records. Until recently, only sodium has been used to qualitatively reconstruct sea ice at glacial-interglacial timescales (e.g. Wolff et al., 2006), but this proxy has limitations at annual and decadal scales, because of the noise input caused by meteorology and open water sources (Abram et al., 2013). Methane sulfonic acid (MSA) is an end product of the oxidation of dimethylsulfide (DMS), which is produced by phytoplankton. MSA deposition has been successfully linked to Antarctic winter sea ice extent (Curran et al., 2003; Abram et al., 2010) on decadal to centennial scales. Yet, the correlation between MSA records and satellite sea ice observations have been shown to be strongly site dependent (Röthlisberger et al., 2009). Post-depositional processes causing loss and migration in the ice layers have also been widely reported to affect MSA, especially in Greenland and at low accumulation sites (Delmas et al., 2003; Weller et al., 2004; Isaksson et al., 2005; Abram et al., 2008).

Victoria Land has been intensively studied for the past two decades. The Taylor Dome (Grootes et al., 2001) and Talos Dome (Stenni et al., 2011) deep ice cores respectively provide 150 kyr and 300 kyr climatic records directly influenced by marine airmasses. Studies on aeolian mineral dust concentration (Delmonte et al., 2010; 2013, Albani et al.,2012) and elemental composition (Vallelonga et al., 2013, Baccolo et al.,2016) were carried out in Talos Dome to understand dust deposition and provenance to Victoria Land.
Albani et al. (2008) pointed out the presence of marine compounds (ikaite) at Talos Dome, typically formed at the early stages of sea ice formation. Back trajectory calculations show that favourable events for air mass advection from the sea ice surface to Talos Dome are rare but likely to occur.
Within the framework of the ITASE program (International Trans-Antarctic Scientific Expedition, Mayewski et al., 2005), several traverses were carried out to evaluate the spatial patterns of isotopic values and chemical species (Magand et al., 2004; Proposito et al., 2002; Becagli et al., 2004, 2005; Benassai et al., 2005).

We present here bromine, iodine and sodium deposition in coastal East Antarctica, by investigating their total concentrations within a series of shallow firn cores, covering the 2010-2013 time period. The cores





were drilled during a traverse performed in late December 2013 in Victoria Land (East Antarctica), from Talos Dome (72°48' S, 159°06' E) to GV7 (70°41' S, 158°51' E). The variability of these species at sub-annual timescales will inform on timing and seasonality as well as spatial patterns of their deposition. Such information is necessary for the interpretation at longer timescales of these elements and possible depositional or post depositional effects. These sub-annual resolution investigations are still limited to the Indian sector (Law Dome - Spolaor et al., 2014) and the Atlantic sector (Neumayer station - Frieß et al., 2010) of Antarctica. This study will test the regional variability of these tracers, providing measurements from the Ross Sea to the Indian sector that remains otherwise unstudied.

## 2. Sampling and analyses

### 2.1 Traverse sampling

The traverse was performed in the northern Victoria Land region of East Antarctica (Fig. 1) from the 20th November 2013 to the 8th January 2014. The starting and ending locations were Talos Dome (72°48'S, 159°12' E) and location '6' (see Fig. 1), close to GV7 (70°41' S, 158°51' E), for a total distance of about 300 kilometers. Talos Dome (275 km WNW from Mario Zucchelli station) is located approximately 250 km from the Ross Sea and 290 km from the Indian Ocean. GV7 is a peripheral site on the ice divide coming from Talos Dome, located just 95 km from the Indian Ocean.

During the transect, seven shallow cores, labelled hereafter TD (Talos Dome), 10, 9, GV7, 8, 7 and 6 were hand drilled to 2 -m depth (except for GV7 which was 2.5 m). The main characteristics of the coring sites are reported in Table 1. Density profiles were obtained from each core immediately after drilling.

The hand auger had a diameter of 10 cm and consisted of an aluminum barrel equipped with fiberglass extensions. The cores were sampled in the cold laboratory at Cà Foscari University of Venice under a class-100 laminar flow hood. Each core was cut with a commercial hand saw and decontaminated through mechanical chiseling by removing approximately 1 cm of the external layer. Every tool was cleaned repeatedly with Ultrapure water. The cores were then subsampled at 5 cm resolution (3 cm for the GV7 core) into polyethylene vials previously cleaned with UPW and then kept frozen at -20 °C until analysis.

### 2.2 Analytical measurements

Total sodium (Na), bromine (Br) and iodine (I) concentrations were determined by Inductively Coupled Plasma - Sector Field Mass Spectrometry (ICP-SFMS Element2, ThermoFischer, Bremen, Germany) at Cà Foscari University of Venice. For a detailed description of the analytical method, see Spolaor et al., 2013a. Each analytical run was started and ended by a UPW cleaning session of 3 min to ensure a stable background level throughout the analysis.

The external standards that were used to calibrate the analytes were prepared by diluting a 1000 ppm stock IC solution (TraceCERT® purity grade, Sigma-Aldrich, MO, USA). The standard concentrations ranged between 10 and 4000 ppt. The calibration regression lines showed correlation coefficients $R^2$>0.99 (N=6, p=0.05). The detection limits, calculated as three times the standard deviation of the blanks, were 50 and 5 ppt for bromine and iodine respectively and 0.8 ppb for sodium.
Procedural UPW blanks were analyzed periodically to test the cleanliness of the instrument lines. Precautions were also taken to minimize the exposure of the samples to direct light, in order to minimize bromine and iodine photolysis reactions.

Stable isotopes of water ($^{18}$O and D) measurements were conducted on aliquots of the same samples at the Center for Ice and Climate (Copenhagen, Denmark) using a Cavity Ring-Down Spectrometer (Picarro, Santa Clara, USA) using the method described by Gkinis et al. (2010). Septum-sealed glass vials were used for these measurements to prevent any sample evaporation during the experimental phases.



## 3. Results and discussion

### 3.1 Stable water isotopes and snow accumulation

The cores were dated based on the seasonal variations identified in the stable water isotopic profiles (both $\delta^{18}O$ and $\delta D$). Isotope ratio minima (representing midwinter) can be easily identified (Fig. 2). Almost all of the cores cover the period between 2010 and late 2013, providing four years of snow deposition. The only exception is represented by core 6, whose upper layer is missing.

The annual deposition signal looks less clear in the two cores that were drilled at the sites with the highest elevation and the closest to the Ross Sea, cores TD and 10, and especially for 2013 in core 10. The two sites are probably the most affected by surface remobilization and isotopic diffusion due to low accumulation. Indeed, non-uniformities in the shallow snow layers such as sastrugi, dunes, wind crusts and other features have been identified as an important aspect of the surface morphology around the Talos Dome area (Frezzotti et al., 2004; 2007).

The annual accumulation rates were calculated by selecting the depth intervals included within consecutive maximum or minimum $\delta^{18}O$ values (Table 2) and accounting for the measured density profiles. Table 2 also includes accumulation rates in Victoria Land reported from previous studies. The GV5 site is located between sites 10 and 9 (Fig. 1).

The accumulation rates found during the traverse are in general agreement with the previous works (Becagli et al., 2004; Frezzotti et al., 2007), except for Talos Dome. The accumulation values calculated from the smoothed isotopic profile in Talos Dome are well above those measured by the stake farm (n=41, C. Scarchilli, *personal communication*) for the same years. The fluxes of deposition of sodium, bromine and iodine along the transect are calculated using accumulation rates from this work, except for Talos Dome, where we used the stake farm values.

The accumulation pattern along the transect increases from Talos Dome to the Southern Ocean (GV7, 8, 7, 6), as the previous works have also found (Magand et al., 2004; Frezzotti et al., 2007). Scarchilli et al. (2010) already pointed out how Talos Dome receives 50% of its total precipitation from the north-west (Indian Ocean), 30% from the east (Ross Sea and Pacific Ocean) and approximately 15% from the interior of the plateau. In this picture, our accumulation data show a decrease from the Indian Ocean moving away from the Indian Ocean coasts and approaching Talos Dome.

The sites are located at decreasing altitudes moving from Talos Dome site (highest point) towards the coast facing the Indian sector (site 6). The minimum $\delta^{18}O$ value found in each core shows a decreasing trend with altitude, with an elevation gradient of -1.35 ‰(100m)$^{-1}$. This super-adiabatic lapse rate is confirmed by the surface snow samples collected taken during the 2001/02 ITASE traverse (Magand et al., 2004).

### 3.2 Sodium, Bromine and Iodine

The main statistical parameters of the measured samples are summarized in Table 3. Sodium shows a mean concentration of 34 ppb, in agreement with published values in this area (Becagli et al., 2004, Bertler et al., 2005, Severi et al., 2009). Among the three elements, sodium shows the highest standard deviation (61%) because of the high variability of sea spray inputs at coastal sites. Singularities up to 200 ppb are probably associated to sea salt rich marine storms. Iodine has an average concentration of 43 ppt, associated with a lower variability compared to bromine and sodium.

The bromine enrichment has been calculated as the bromine excess with respect to sea water concentrations, $Brenr = [Br]/(0.006 \cdot [Na])$, where [Br] and [Na] are the bromine and sodium concentrations in the sample and 0.006 is the bromine-to-sodium concentration ratio in sea water (Millero, 2008). Benassai et al. (2005) have concluded that sea-salt sodium is the dominant fraction (more than 80%) of the total sodium budget in this area. No correction to sodium was therefore applied for this calculation. Despite bromine being a sea salt



marker like sodium, it is recycled over halogen rich sea ice surfaces (i.e. first year sea ice, FYSI) and suspended
sea salt aerosol, and exponentially released as $Br_2$, Eq. (1):

$$HOBr + HBr \rightarrow Br_2 + H_2O \qquad\qquad (1)$$

Therefore, bromine enrichment signals sea ice presence, on the hypothesis that sodium is left unchanged from
emission to deposition. Bromine enrichment has already been linked to sea ice presence in both Arctic and
Antarctic coastal sites (Spolaor et al., 2013b, 2014, 2016; Vallelonga et al., 2016).
The entire set of bromine enrichment values is visible in Fig. 3. The values extend from a minimum of 0.5 to
17, with an average of $4.6 \pm 0.1$. Notably, more than 98% of the samples show values greater than 1 (i.e. sea
water value). A detailed insight on the few <1 values revealed that these samples are associated with very high
contributions of sodium inputs (>120 ppb), therefore likely associated to strong marine events. Such
distribution of enrichment supports the theory that this parameter is, in these coastal sites, a marker of sea salt
aerosol with an extra contribution from sea ice.
Figures 4-5-6 and 7 show the variabilities of $\delta^{18}O$ (upper plot), sodium (middle plot, left axis), bromine (middle
plot, right axis), iodine (lower plot, right axis) and bromine enrichment (lower plot, left axis) on an age scale
for the different coring sites. Thick lines represent 3-month running means of the raw data (circles).
Sodium timeseries show great variability: peaks are often found in summer, although they are also observed
in winter in cores TD and 8. These findings confirm that, as previous works pointed out (Curran et. al., 1998),
in coastal sites storm events carrying open ocean sea salts are more important than sea ice as a sea salt source,
although the high level of variability suggests also that meteorology and natural variability play a role. Bromine
and bromine enrichment show annual variations, with maximum values in during late spring-summer. Iodine
shows a more stable signal throughout the year and high winter singularities in cores TD, GV7 and 8.
The timing of the bromine enrichment signal in ice cores relies on the combined effect of sea ice and sunlight,
responsible for the photochemical production and release of molecular bromine, $Br_2$ (Pratt et al., 2013). Sea
ice area in the 130-190° E sector was calculated for the 2010-2013 period using web available NSIDC passive
microwave sea ice concentration data (Meier et al., 2013). Figures 8 and 9 show the minimum and maximum,
found in January 2010 and August 2011, respectively. The monthly sea ice areas from 2010 to 2013 were
calculated for such sector and plotted in Fig. 9a (blue); each monthly value was normalized to the total annual
sea ice area. The minimum sea ice is found in February, while a longer lasting maximum throughout winter
and spring is observed, before a rapid decrease from November. Monthly insolation values were recovered
from the Japanese Syowa Station (69° S, 39° E), Fig. 9a (red points).
The sub annual distribution of bromine enrichment along the transect is shown in Fig. 9b (blue). Each bins
contains the cumulative monthly value for every year in every core, normalized by the total value of each year
(which may change according to year and location). The histogram is then normalized by the overall sum
measured in the transect. The distribution shows a clear sub-annual oscillation with lowest and highest annual
contribution in May (late autumn) and November (late spring), respectively. The combined effect of sea ice
and insolation (Fig. 9b, magenta product distribution) shows the same features, with maximum in spring. Such
comparison demonstrates the dependency of bromine enrichment on the combined effect of sea ice and
insolation. Monthly sea ice area values are reported in Fig. 9c (blue), together with annual averaged values of
bromine enrichment (black) and first year sea ice, FYSI (red), calculated as the difference of maximum and
minimum sea ice area. A coherent trend in both $Br_{enr}$ and FYSI is observed for the first three years, and not for
the last one, although uncertainties cannot exclude a consistency over the whole period. A longer record would
be needed to evaluate the correlation and quantitatively reconstruct past sea ice in this sector.
Table 4 shows the average annual iodine concentrations for each location, together with its standard deviation.
The mean values (0.043 ppb on average) are close to the background values found in Antarctic shallow firn
cores near the research stations of Neumayer (Frieß et al., 2010) and Casey (Law Dome, Spolaor et al., 2014)
respectively. Unlike previous observations of a clear winter peak of iodine with concentrations up to 0.6 ppb
(Neumayer) and 0.3 ppb (Law Dome), no clear seasonality is observed for the transect samples, with annual





variability around 10-15%. Core 7 (Fig. 6) shows some variability which corresponds to winter peaks. High
iodine concentrations are observed in core 8 during the 2012 winter, in association to a strong sea salt (sodium)
input, although similar strong winter peaks are observed in 2011 at GV7 and TD sites.
The low background level and low variability of iodine found along the transect reflect a low input of iodine
in this area of Antarctica compared to other locations. This picture is confirmed by tropospheric measurements
of IO from satellites (Fig. 10, right panel), which show IO concentrations close to detection limit over the area
of the transect compared to Law Dome, Neumayer, or any other coastal location. The high elevation of the
traverse area, compared to the others is likely to play a role in preventing efficient iodine transport from the
source areas.
Frieß et al. (2010) and Spolaor et al. (2014) have attributed iodine seasonal signal pattern to summertime
photochemical recycling of IO from the snowpack, leading to depletion in the summer layers and higher
concentrations in winter when the polar night starts. The lower variability found across the Northern Victoria
Land traverse cores could result from a reduced summer recycling due to low iodine concentrations available
the snow.

### 3.3 Spatial flux variability

Glaciochemistry around Antarctica is very strongly influenced, among several properties, by the distance from
the sea and the pathways of the air mass trajectories (Bertler et al., 2005). Atmospheric circulation patterns
around the Talos Dome area have been studied by Scarchilli et al. (2010), who have shown that the main input
is represented by the Southern Ocean (Indian sector) with a lower contribution from the Ross Sea.
The spatial variability of sodium, bromine and iodine is investigated in Fig. 11. The twelve panels show the
mean annual fluxes for each core in relation to the distance from the Indian Ocean. Sodium fluxes show the
highest values and variability around the closest locations to the Southern Ocean (GV7, 8, 7, 6), where the
accumulation increases. After rapidly decreasing within the first 100 km, the sodium flux becomes stable, as
the input from the SO decreases but the one from the Ross sea gradually increases. Talos Dome, the closest
location to the Ross sea (290 km from the Indian Ocean – point on the furthest right in every plot of Fig. 11)
shows an average annual sodium flux of $(2400 \pm 970)$ µg m$^{-2}$ yr$^{-1}$, consistent, within uncertainties, with the
value of $(1500 \pm 500)$ µg m$^{-2}$ yr$^{-1}$ found by Spolaor et al. (2013b). Bromine exhibits a similar behavior to
sodium, with a homogeneous flux within cores TD, 10 and 9 and an increase (up to 3 times) in the last 100
km from the SO. Elevation could partly account for the fractionation of sodium and bromine, having the 180
m of height difference separating GV7,8,7 and 6, and 360 m from GV7 to Talos Dome. The effect of
elevation yet is combined to the influence of the distance from the source in order to resolve the two effects.
A slightly lower fractionation after 100 km from the SO is observed for iodine, confirming the homogeneous
satellite measurements of IO (Fig. 10, right).



# 4. Conclusions

The 2013/14 Talos Dome – GV7 traverse provided an opportunity to expand the existing sodium dataset in Victoria Land and investigate important features of bromine and iodine temporal and spatial variabilities, so far only available in Antarctica at Law Dome and Neumayer station.

The accumulation rates agree with previous studies, with increasing values from the Ross Sea to the Southern Ocean. Accumulation rates calculated for Talos Dome are higher than previously reported, likely caused by isotopic diffusion and remobilization at this site. The locations near the Southern Ocean exhibit high variability due to the higher accumulation.

Sodium and bromine concentrations in the snow samples result in a positive bromine enrichment to seawater, confirming the sea ice influence in the area for the extra bromine deposition. While sodium does not capture clear sub-annual variations associated with sea ice, bromine enrichment shows consistent seasonal variabilities with late spring maxima (November). It is possible to relate such seasonality to the combined effect of sea ice growth and sunlight, which trigger photochemistry above fresh sea ice. The timing of deposition is coherent among Victoria Land, Law Dome (Indian sector) and Neumayer (Atlantic sector). Iodine shows an average value of 0.04 ppb, similar to background values observed in the Antarctic coastal locations of Law Dome and Neumayer. Unlike those locations, low iodine annual variability and no consistent seasonality of the signal are observed in the traverse samples.

The spatial variability study reveals homogeneous fluxes over the transect length, with an increase in absolute values and variability at the sites close to the Indian Ocean, due to high accumulation and proximity to the coasts. Uniform satellite values of BrO and IO over Victoria Land confirm the snow measurements. A fractionation due to distance of these proxies is not found probably due to the combined double input of air masses from the Ross Sea and the Indian Ocean. A transect towards the interior of the plateau would give an insight on this feature.

The trends of annual bromine enrichment and first year sea ice in the 130-190° E sector are comparable, supporting the use of bromine enrichment as FYSI proxy.

# Acknowledgements

We thank the scientists who conducted the traverse and provided the samples, the chemistry group in Venice for the chemical measurements as well as the isotope laboratory in Copenhagen for the measurements of the water isotopes. Thank also to Rasmus Anker Pedersen and Emilie Capron for the useful suggestions and comments.

This research was carried out in the framework of the Project on Glaciology and Paleoclimatology of the Italian PNRA National Antarctic Program.

The research leading to these results has received funding from the European Research Council under the European Community's Seventh Framework Programme (FP7/2007-2013) / ERC grant agreement 610055 as part of the ice2ice project.



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





**Table 1.** Core drilling site information.

| Core Site | Core depth (cm) | Lat. (S) | Long. (E) | Elevation (m a.s.l) | Distance from Ross sea (km) | Distance from Indian Ocean (km) |
|---|---|---|---|---|---|---|
| TD | 200 | 72° 48' | 159° 06' | 2315 | 250 | 290 |
| 10 | 200 | 72° 12' | 158°41' | 2200 | 310 | 240 |
| 9 | 200 | 71° 21' | 158° 23' | 2151 | 380 | 180 |
| GV7 | 250 | 70° 41' | 158° 51' | 1957 | 430 | 95 |
| 8 | 200 | 70° 36' | 158° 35' | 1934 | 440 | 90 |
| 7 | 200 | 70° 31' | 158° 25' | 1894 | 460 | 90 |
| 6 | 200 | 70° 21' | 158° 24' | 1781 | 470 | 85 |




**Table 2.** Summary of accumulation rate data from Northern Victoria Land. All uncertainties (shown in parentheses) are 1σ errors.
(a) this work.
* Uncertain due to smoothed isotopic signal.
(b) Becagli et al., 2004.
(c) Frezzotti et al., 2007.
(d) from stake farm (n=41) (C. Scarchilli, *personal communication*).
(e) 1966-96 (Stenni et al., 2002).

| Core | Accumulation rates (kg m⁻² yr⁻¹) | | | | | | | |
|---|---|---|---|---|---|---|---|---|
| | 2013 traverse [a] | | | | | 2001/02 [b] | 1965-2001 [c] | 2001-2012 [d] |
| | 2013 | 2012 | 2011 | 2010 | Average | | | |
| TD | 223* | 144* | 187* | - | 185 (31) | 104 (37) | 86.6[e] | 71 (4) |
| | - | 66[d] | 107[d] | 78[d] | 81 (17)[d] | | | |
| 10 | 260* | 140 | 140 | 120 | 133 (9) | GV5 156 (27) | GV5 129 (6) | |
| 9 | 180 | 180 | 180 | 180 | 180 (0) | | | |
| GV7 | 228 | 261 | 260 | 156 | 232 (32) | 261 (50) | 241 (12) | |
| 8 | 240 | 260 | 280 | - | 260 (16) | | | |
| 7 | 220 | 180 | 200 | 180 | 195 (18) | | | |
| 6 | - | 200 | 260 | 200 | 220 (29) | | | |






**Table 3.** Mean parameters for the 2013/14 Talos Dome – GV7 traverse samples. DL = detection limit, calculated as 3 times the standard deviation of the blanks.

|            | Na (ppb) | I (ppb) | Br (ppb) |
|------------|----------|---------|----------|
| # values   | 373      | 374     | 377      |
| Min        | 9        | 0.03    | 0.2      |
| Max        | 196      | 0.09    | 2.2      |
| Mean       | 34       | 0.043   | 0.7      |
| DL         | 0.8      | 0.005   | 0.05     |
| Std dev %  | 61       | 23      | 42       |



**Table 4**. Iodine average concentrations and variability during the 2013-2010 time period. All values are expressed in ppb.

| Core | 2013 | | 2012 | | 2011 | | 2010 | |
|------|------|---------|------|---------|------|---------|------|---------|
|      | I | St. dev. | I | St. dev. | I | St. dev. | I | St. dev. |
| TD  | 0.047 | 0.004 | 0.044 | 0.002 | 0.048 | 0.013 | - | - |
| 10  | 0.041 | 0.005 | 0.043 | 0.001 | 0.049 | 0.008 | 0.040 | 0.005 |
| 9   | 0.038 | 0.003 | 0.041 | 0.010 | 0.046 | 0.008 | 0.047 | 0.003 |
| GV7 | 0.044 | 0.004 | 0.042 | 0.004 | 0.043 | 0.004 | 0.047 | 0.005 |
| 8   | 0.033 | 0.002 | 0.049 | 0.021 | 0.032 | 0.002 | - | - |
| 7   | 0.038 | 0.006 | 0.034 | 0.004 | 0.037 | 0.009 | 0.041 | 0.008 |
| 6   | - | - | 0.039 | 0.002 | 0.044 | 0.006 | 0.041 | 0.008 |






**Figure 1.** (a) Schematic map of the traverse area and coring sites, marked with stars. The cores were drilled between Nov 20th 2013 and Jan 8th 2014 (early austral summer). (b) Minimum (left, February) and maximum (right, July) sea ice concentrations in 2010 (NSIDC data from Meier et al., 2013).

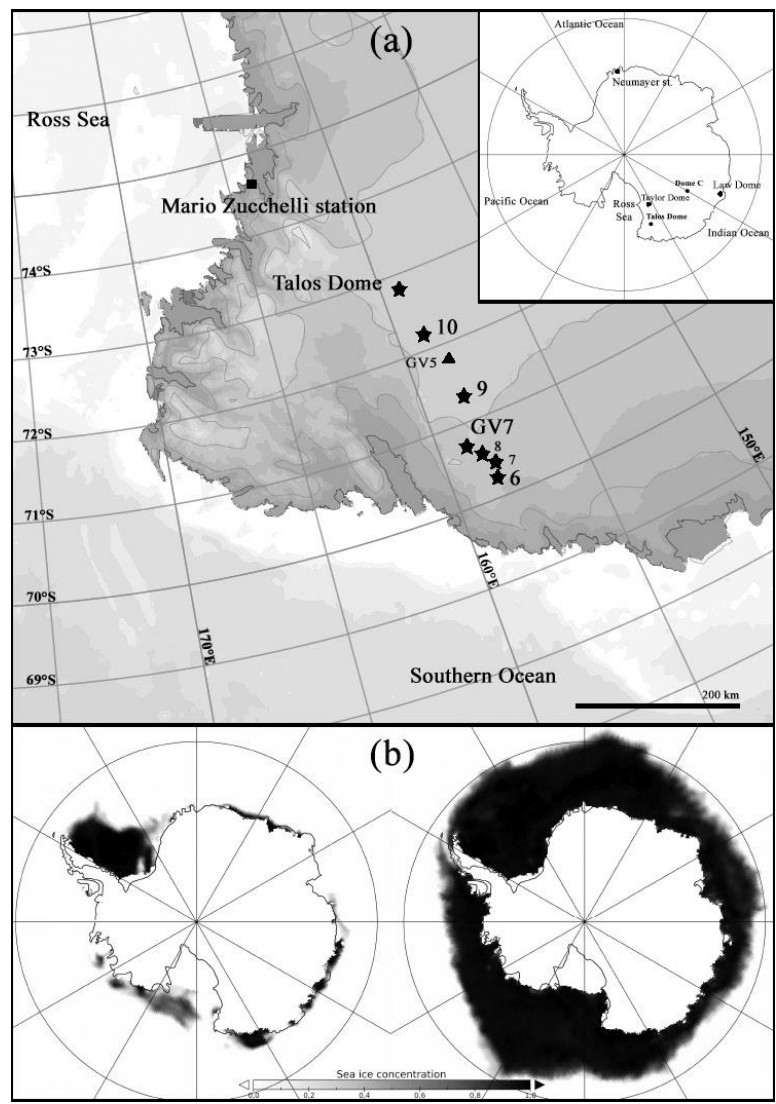



**Figure 2.** δ¹⁸O (thick line) and δD (dashed line) profiles of the cores. Resolution of sampling is 5 cm. The winter of each year is indicated with lines in correspondence with the water isotope minima. Core 10: the 2013 winter layer is uncertain.

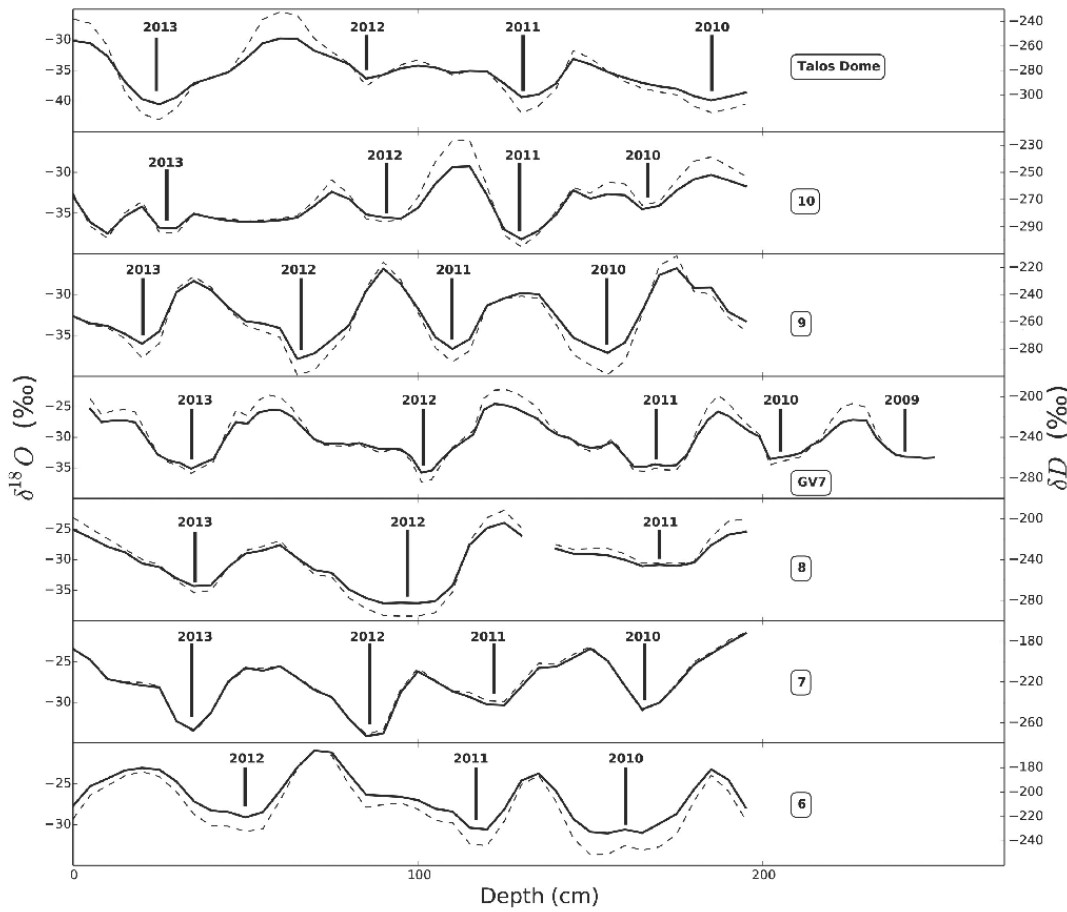




**Figure 3**. Distribution of bromine enrichment values of the entire set of samples. The dashed line indicates the seawater value ($Br_{enr}$
$= 1$).

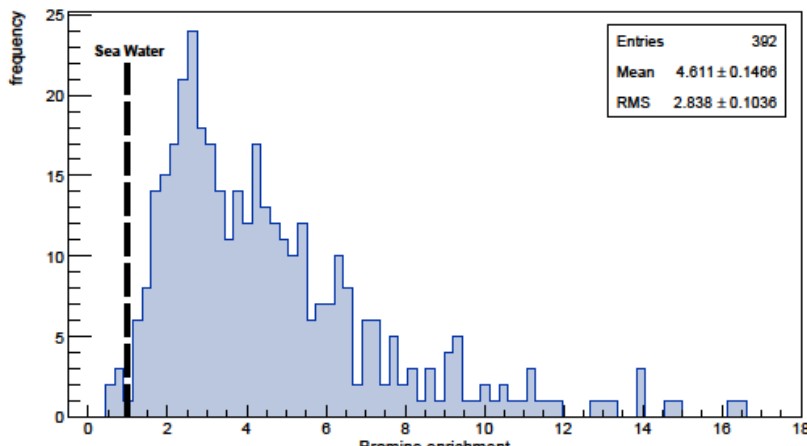




**Figure 4.** $\delta^{18}O$, Na, Br, $Br_{enr}$ and I profiles in cores TD (left) and 10 (right).

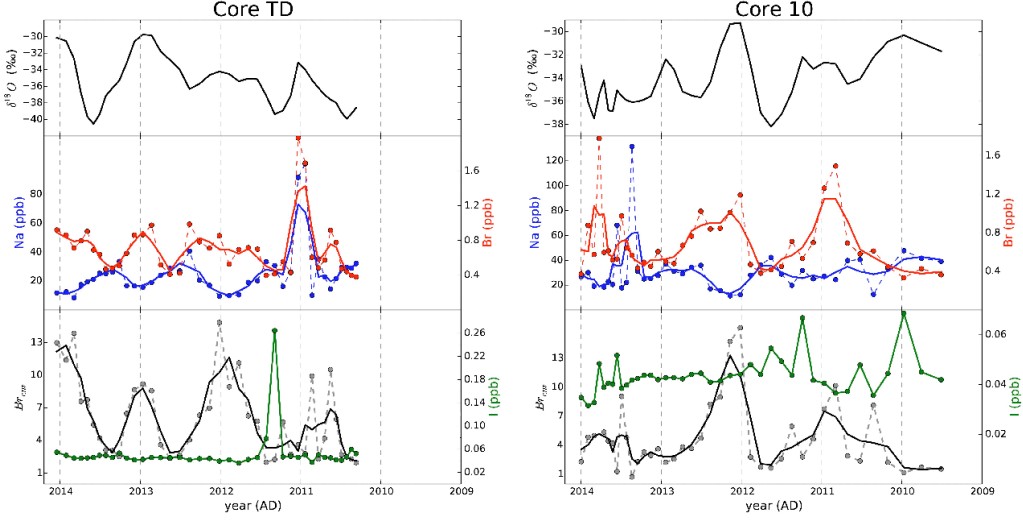






**Figure 5.** $\delta^{18}$O, Na, Br, Br$_{enr}$ and I profiles in cores 9 (left) and GV7 (right).

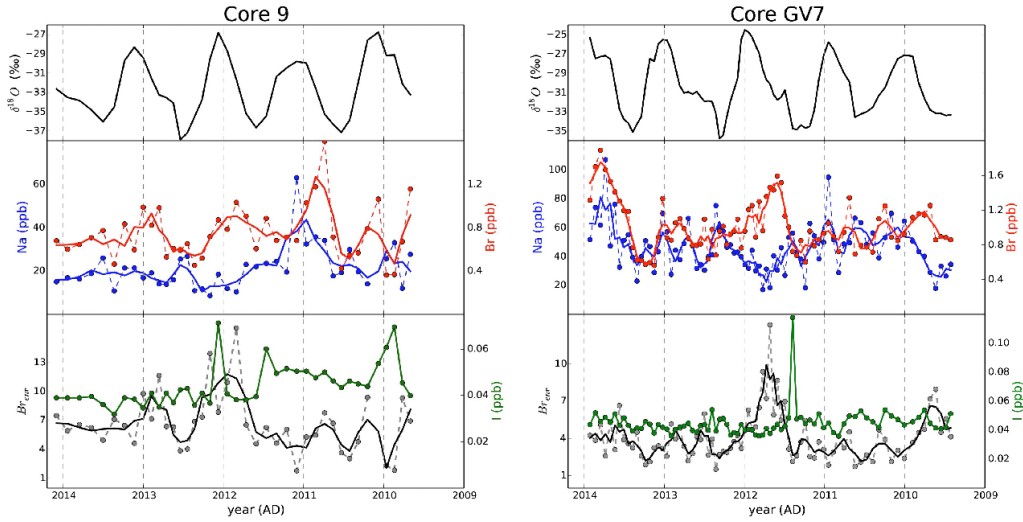






**Figure 6.** $\delta^{18}$O, Na, Br, Br$_{enr}$ and I profiles in cores 8 (left) and 7 (right).

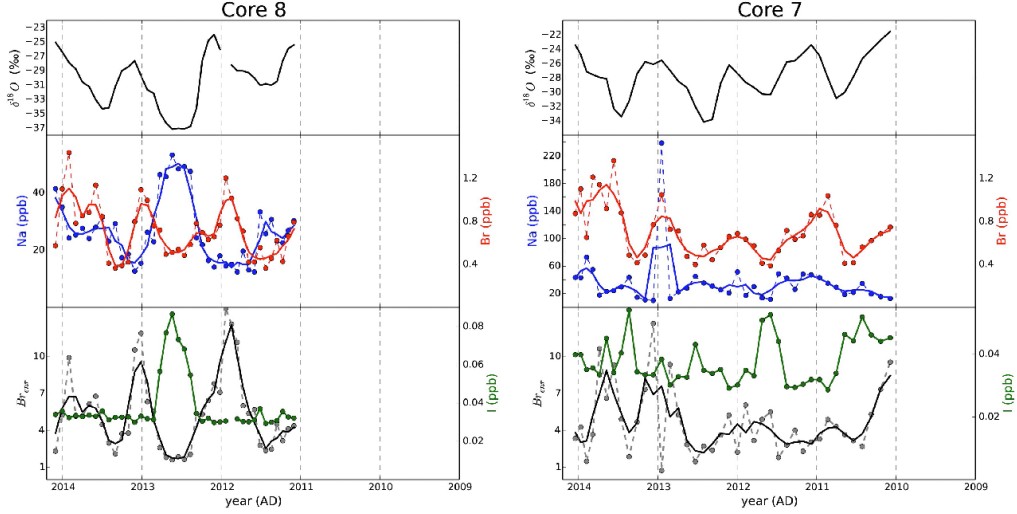


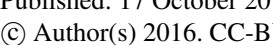


**Figure 7.** $\delta^{18}$O, Na, Br, Br$_{enr}$ and I profiles in core 6.

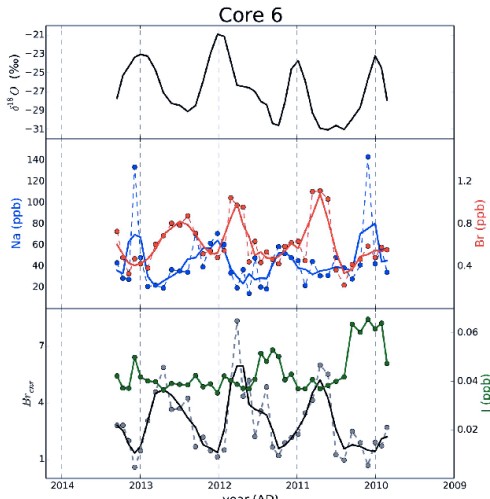




**Figure 8.** Minimum (a) and maximum (b) sea ice in the 2010-2013 period within 130-190° E longitude sector.

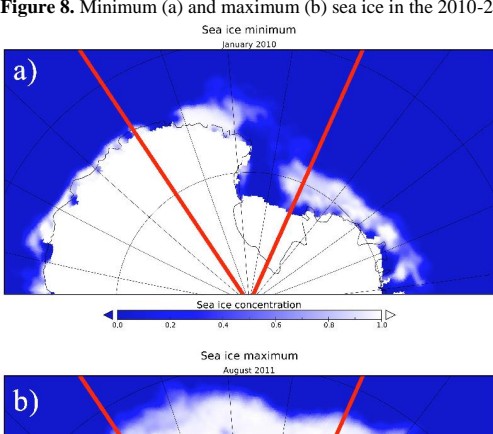

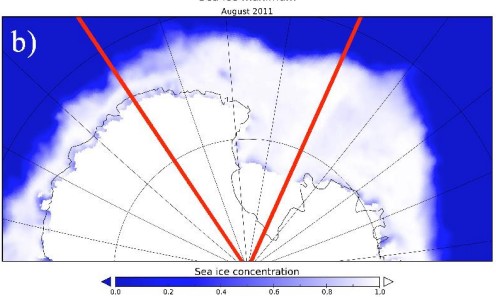





**Figure 9. (a)** Monthly values of sea ice area (blue) within the 130-190° E sector and solar radiation at 69° S (red). **(b)** % of annual bromine enrichment along the traverse, normalized to the total annual amount: the monthly trend shows a seasonal feature with maximum in late Spring (November). The histogram considers every year and every core of the transect. The shaded blue area shows the systematic uncertainty associated to the dating. The error bars represent the normalized rms. The magenta band represents the product distribution of normalized sea ice area and insolation, expressed in annual percentage. **(c)** Monthly sea ice area values (blue) from 2010 to 2013, with annual values of FYSI (red) and averaged bromine enrichment (black).

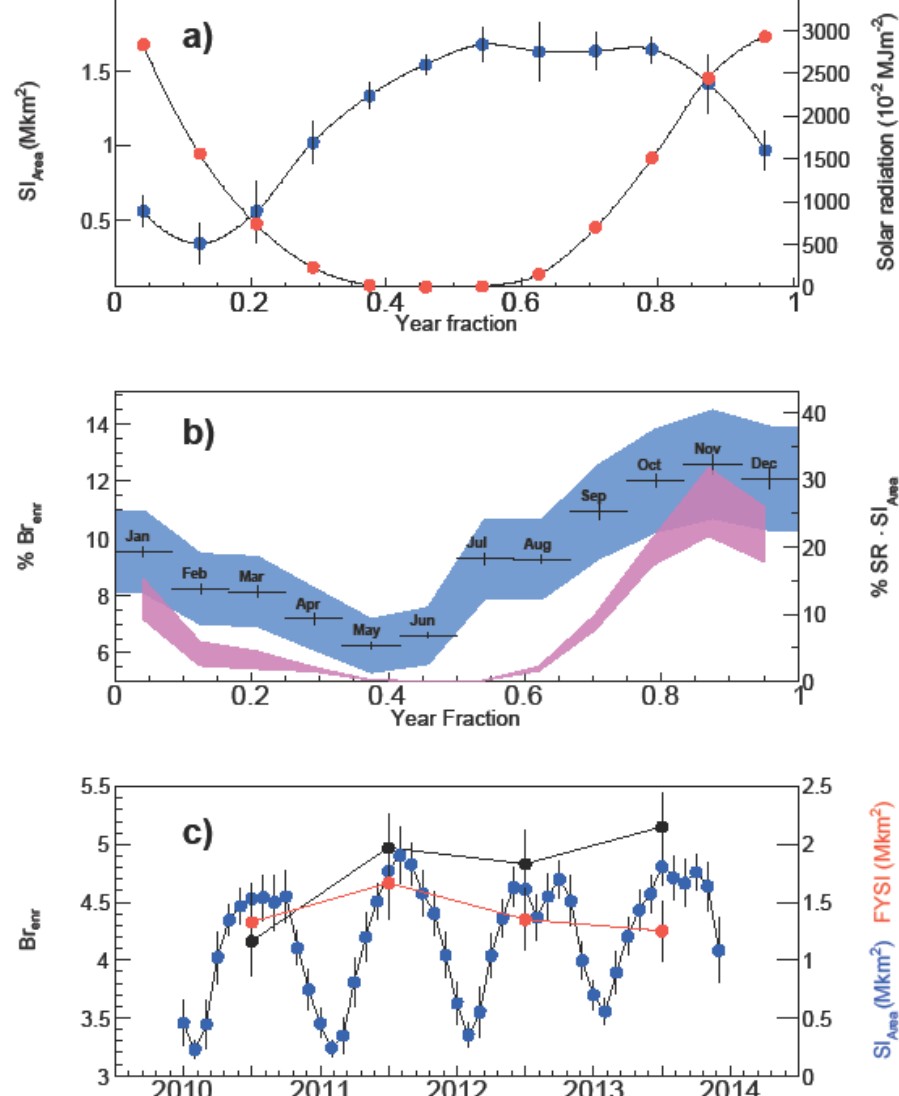



**Figure 10.** Average atmospheric column concentrations of BrO and IO in Antarctica between 2009 and 2011, from Spolaor et al., 2014.

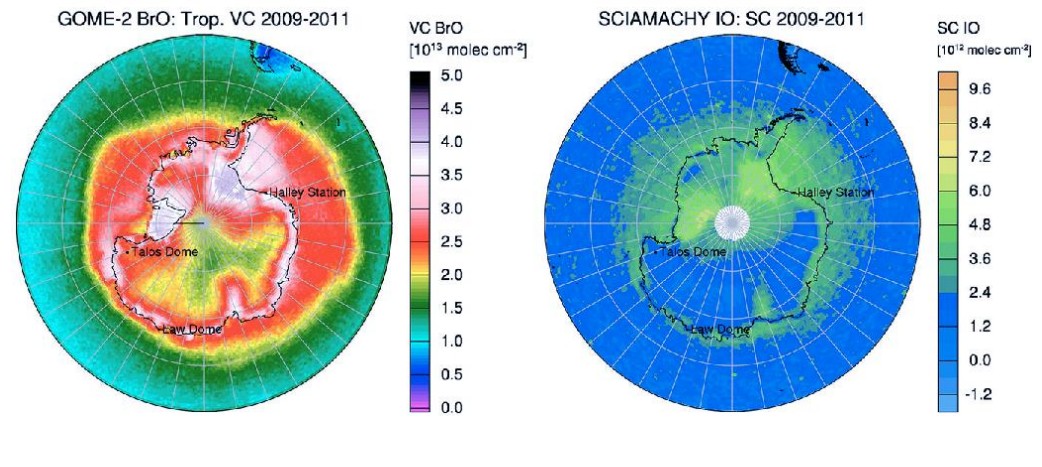



**Figure 11.** Mean annual fluxes of sodium (blue), bromine (red) and iodine (green) as a function of distance from the Indian Ocean. Each dot represents a location along the traverse.

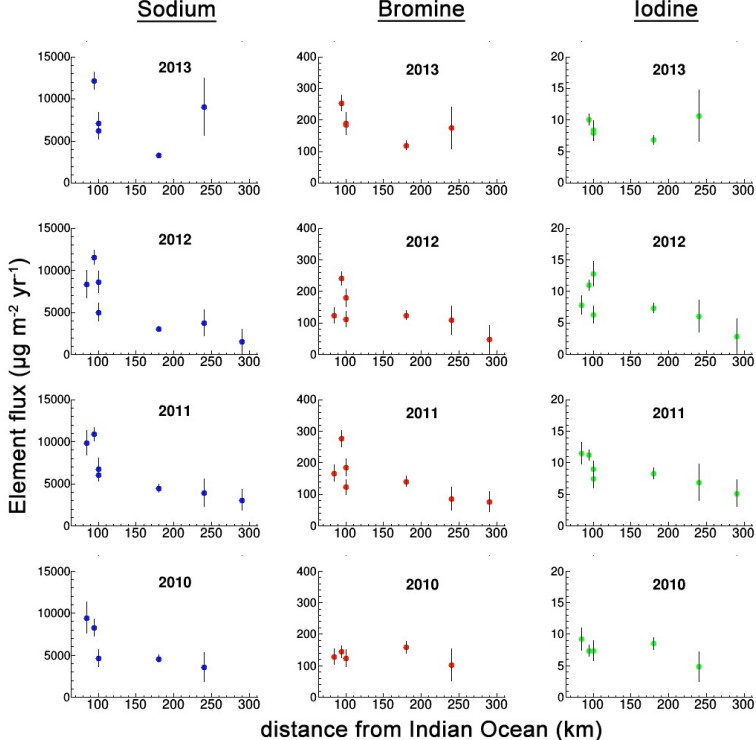