# Peer review of "Bromine, iodine and sodium in surface snow along the 2013 Talos"

_The Cryosphere, 2016_

## Referee Comment (RC1) · Anonymous Referee #1 · 23 Nov 2016

It is well-established that the sea ice zone is a hotspot for Br (and with less certainty, I) chemistry. This has been shown by the high concentrations of BrO observed from satellite data in spring, and from ground-based data. The mechanism is understood to be that inorganic Br is activated from salty material (sea ice or aerosol). With this in mind, Spolaor et al (2013b in the present manuscript) found that Br was enriched in interglacial ice and depleted in glacial ice at Talos Dome. They proposed that this resulted from the halogen chemistry over sea ice, and that the enrichment or depletion might be used as an index of sea ice extent in the past. Their mechanism relied on their suggestion that Br/Na would reduce with distance from the sea ice edge, because the enriched material (as gas phase HBr) was deposited faster than the depleted (NaBr)

[Figure]

sea salt aerosol. This was a surprising suggestion because previous work (Simpson et al, 2005) had suggested exactly the opposite (for the Arctic): that inland snow would be enhanced due to a longer lifetime of HBr.

Clearly it is impossible to consider Br as a sea ice proxy until at least a reasonable understanding of the mechanism leading to temporal changes is understood, so studies attempting to elucidate this are very welcome. The present paper is aimed at doing this, by making a spatial transect (including seasonal information) of Na, Br and I in a part of East Antarctica. Unfortunately the location of the traverse is particularly badly chosen for such a study, because the sites sit between two marine areas, the Ross Sea and the main Southern Ocean. This has the effect that sites that are further from one potential source are nearer to the other, so that even if the data showed very clear trends, opposing interpretations would have been possible. As it happens there is little clarity in the data, which is not entirely the authors' fault but does mean that this is a paper which advances knowledge only incrementally. It is probably justified to consider publishing it after significant changes have been made, if only to indicate the complexity of the problem, and to show how premature it is to consider Br as a sea ice proxy until far more detailed and well-designed experiments and sampling campaigns have been carried out. The paper itself is relatively short but with a very high number of tables and figures that really don't add to it, so among other things I would recommend losing some of these in the next version. There are also some unjustified interpretations (such as that in the abstract regarding Fig 9), which definitely must be modified.

Comments:

Abstract line 27-28. The last sentence is not justified. This is based on Fig 9c, with 4 data points. No statistics are given but I see no correlation at all, and a rough attempt to plot the data gave an rˆ2 very close to zero, utterly insignificant.

Sections 1 and 2 are generally OK, with two minor comments:

Line 36, remove the word layer. There is a "layer" of ozone in the stratosphere but there

is no layer in the troposphere.

Line 68. The Rothlisberger article in a newsletter is not a good reference here. Better would be reference 3 or one of the other papers by Abram.

Line 162-167. While I agree that, taking old and new data into account, there is a tendency (as one would expect) of lower accumulation rates as we move further from the ocean, obviously this is somewhat undermined by the high value of 185 kg mˆ-2 aˆ-1 at TD. In the table the authors try to mitigate this by putting asterisks on "uncertain years", saying that their value is uncertain because the isotope signal is less clear. I don't think Fig 2 really justifies this – the least clear assignment at TD (2012) is at least as obvious as the one for 2011 at site 8 for example, and yet this has no asterisk against it in Table 2. I think a better way to handle this would be to remove the asterisks from the table, but to say that the inconsistency between the accumulation rate derived from the core at TD and that derived from the stake farm and previous measurements suggests that the isotopic assignments of years may be incorrect at TD, and that the profile contains more years than have been assigned.

Table 3 is unnecessary. Most of the information is anyway given in the text, but anyway a table like this that mixes different sites has no value that I can see. The table should be removed.

Line 189. This sentence is not really correct. Either enrichment or depletion can indicate that the reactions, believed to be focussed on sea ice, have taken place (not just enrichment). In fact the reaction (1) as shown leads only to a depletion of Br from the sea ice. It is only if the Br2 is eventually converted back to HBr that enrichment can occur, if more of this end product gas phase HBr is deposited nullifying the depletion in the aerosol phase. In addition, there are other ways to get such enrichment, as we know from the case of Cl, which is enriched or depleted compared to sodium due simply to production of HCl from the reaction between sulfuric or nitric acid with sea salt (e.g. $H_2SO_4 + 2NaCl \rightarrow Na_2SO_4 + 2\ HCl$). A sentence that would be defendible

would be "Therefore sea ice presence should lead to Br enrichment (after conversion of activated gas phase Br back to HBr) or depletion, depending whether deposition is dominated by the depleted sea salt aerosol or by the enriched gas phase HBr."

Fig. 3 is OK, but I would have found it more useful if you had shown the distribution for individual sites or groups of sites. As a suggestion, you could show one distribution for sites GV7/8/7/6 (less than 100 km from ocean) and another distribution for the other sites.

Fig 4, TD plot is obviously a problem, since you seem to believe that (Table 2) the years may be misassigned. This undermines your interpretations for this site. I'm afraid you can't have it both ways – either the TD accumulation is really high in which case the statements made about how accumulation varies with distance are not supported, or it is not in which case the year assignments shown in the TD section of this figure are wrong.

Fig 8 is really confusing because the map is upside down compared to that used in Fig. 1. In any case a similar figure is shown in Fig 1b. therefore please remove Fig 8, simply incorporating the additional information (basically the red box showing the 130-190E band) into Fig 1b.

Line 215 – why is insolation from a site on the opposite side of Antarctica shown here? It is not even at the latitude of the sites here. Furthermore total solar radiation is very unlikely to be relevant, rather it is the radiation at the UV or near-UV wavelengths that might promote the relevant photochemistry. I suggest using a code such as TUV to calculate available radiation at relevant wavelengths.

Line 205, and Fig 9b. It's hard from Figs 4-7 to really see the seasonality, which you describe as max in late spring/summer for Br. Since you must have calculated it to get to Fig 9b, please add a plot of this sort for each site so we can see how consistent the seasonality is at different sites. This could be shown as individual lines underlying the error band in Fig 9b for example, or as a separate panel if this is less confusing.

Line 221-3. The comparison of the seasonality with the product of radiation and sea ice extent is interesting but it does not "demonstrate" the dependency of Br enrichment on their combined effect, rather it is "consistent with" the idea that there is such a dependency. Please adjust the text.

Lines 225-7. As in the abstract, the correct characterisation of Fig 9c is that, with only 4 years of data, no relationship can be discerned. It is not scientific to say that there is a relationship for 3 years and not for the fourth – statistically this means there is no relationship. You just have to admit that there are not enough data here to know whether Br_enr can be used as an indicator of FYSI. In fact (and related to the next comment) nothing you have written here says why you would expect the enrichment to be related to the FYSI area; what mechanism are you envisaging? If, as in the earlier Spolaor paper, it is via distance from the ice edge (through whatever process) then this should be somewhat reflected in the difference between sites, hence my next comment.

Section 3.3 and Fig. 11. I agree that the Br pattern looks similar to the Na pattern. But this then begs the question, what is the pattern of Br_enr. My impression from this figure and the data in Figs 4-7 is that Br enr is probably rather flat with distance. Since this was the issue that Spolaor suggested controlled the TD glacial-interglacial change, it deserves to be shown and discussed in that context. Please add a discussion.

---

## Referee Comment (RC2) · Anonymous Referee #2 · 8 Dec 2016

Review of "Bromine, iodine and sodium in surface snow along the 2013 Talos Dome – GV7 traverse (Northern Victoria Land, East Antarctica) by Maffezzoli et al.

This paper reports measurements of sodium, bromine, and iodine made in shallow cores in East Antarctica. The data are examined to assess, primarily, bromine enhancement, and potential as sea ice proxy. The subject area is very important and the data are interesting. Overall, however, I find that the paper is overly superficial, and lacks necessary supporting information, including references, to substantiate the arguments presented here.

Major concerns:

**Analytical information** is lacking. The authors present very few analytical details, instead refering readers to the Spolaor et al, 2013a paper. However, some information will be specific to the Maffezzoli paper and should be included, and some additional information would be useful. For example, what was the residual standard deviation for Br, I, and Na for the current work? How were the standards prepared – gravimetric or volumetric methods? Standard concentrations ranged between 10 and 4000 ppt, but for for which species? Spoloar 2013 refers to iodine and bromine being calibrated in this way – but what standards were used for sodium? Presumably halogen standards were separate from sodium standards – did this introduce any uncertainties into the analyses? I am not so familiar with this technique – does it analyse Na, I, and Br in a single run? If not, what uncertainties does this introduce in terms of instrument drift etc? Does the method use columns to separate out the elements of interest? If so, what columns were used? Was any reference material used to really pin down the analytical technique, given the extremely low concentrations being measured? Exposure of samples to direct light was minimised, but how long were they actually exposed for? How long were the samples melted for before they were analysed? Were there any repeat analyses of samples carried out? How much liquid water was needed for each analysis? Presumably the samples were re-frozen before being shipped to Copenhagen for isotopic analysis. Is this likely to have introduced any problems/uncertainties into the analyses? Finally, the manuscript states that during sampling, every tool was repeatedly cleaned with ultrapure water – how many times is "repeatedly". What effort was made to ensure that the tools were clean?

**Dating**: The conclusions drawn in the paper rely heavily on correct dating of the cores, i.e. correctly allocating samples to specific seasons. This is done using stable water isotopes, and the manuscript states (line 140) that "isotope ratio minima (representing mid-winter) can be easily identified". But the authors must give the criteria used for such identification, if other than just by eye. As the authors state, there is clearly a question over assignment of the 2013 mid-winter – how did the authors select the one chosen? I also wonder what happened to the top layer of core 6..? Was it damaged during sampling? The accumulation rate of core 10 is less than core 9, so one would expect the snow at 2m depth to be older in core 10 than in core 9; however, this is not the case as presented in the paper. It makes me wonder whether mid-winter assignments in core 10 are out by a year? Certainly, more information on the criteria used to allocate winter minima must be given for cases where it is not completely clear.

Please explain fully how accumulation rates were calculated, rather than just saying (line 153) that measured density profiles were accounted for – explain the method.

Was only 1 core taken in each site? How do you account for variability between cores drilled at the same site? If this is not considered important, then please say so and explain why.

The authors must provide justification on why they chose to use the stake farm data for Talos Dome over their own method. It is not enough to reject your method, which is used on all other sample sites, just because it does not agree with stake farm data at TD. This selection raises concerns over the validity of the method used for all the other sites.

**Supporting information/background text**: Contextual information is not of sufficiently high standard for The Cryosphere. Information provided is at time simplistic, and at others erroneous. Referencing is poor and must be improved. Many statements are unsubstantiated.

**Below are more minor comments:**

- Line 16: halogen chemistry does not only occur through release of sea salt rich aerosols; various saline condensed phases have been suggested;

- line 18: the statement "halogen species in polar snow samples are shown to be closely related to sea ice extent" is too strong – there is clearly a link to sea ice, but this is far from quantified, so how close it is related is not yet known.

-line 25: the transect revealed homogeneous fluxes" – what type of fluxes? Air-to-snow..? Snow-to-air..?

-line 27: "flux measurements are consistent with the uniform values of BrO and IO"... uniform in time or space..?

Line 36 – there is no tropospheric ozone layer; remove the word "layer"

Line 36 – there are many other papers that should be referenced here other than Barrie et al. There are also some excellent reviews, e.g. Simpson et al. 2007,  Abbatt et al. 2012, that should also be included here.

Line 38 – explain why young sea ice surfaces have high salinity, and how they are thus a source of bromine compounds (presumably you mean to the atmosphere). Also, what are "bromine halides"..??

Line 43 – include the reactions and improve the wording here

Line 45- MAX-DOAS generally refers to a ground-based instrument technique; SCIAMACHY is a satellite-borne uv-vis-nir spectrometer that quantifies BrO and IO columns using the DOAS technique. Please re-word. Also state which satellite instrument you refer to.

Line 47 – "from halogen-rich condensed phases (e.g. sea salt aerosol)" – there are likely to be others as well.

Line 49: Such reactions ... lead to enhancement of bromine in the deposition in the surface snowpack  - give a reference to support this statement.

Line 51 – is the Spolaor reference 2016 a or b?

Line 52 – the Vogt paper is a modelling study, so does not include primary information on iodine sources

Line 66 – DMS is produced from DMSP, not directly from phytoplankton

Line 70- I believe Mulvaney and Pasteur were the first to report post-depositional movement of MSA in ice cores.

Line 78 – "Back trajectory calculations show that favourable events of air mass advection from the sea ice surface to TD are rare but likely to occur" – what calculations are these? Which model? Did the authors run them, or are they in a published paper? If the latter, then give references. If the former, give more information. An example plot would be useful with some sort of analysis of air mass origin.

Actually the point being made in paragraph of line 72 to line 82 is not clear.

Line 90 and 93 – "Indian **ocean** sector"

Line 92 – Friess et al did not report Br in their snow pit.

Line 104 – it would help the reader to give the distance between the cores.

Line 184 – The description of bromide release needs improving. e.g."bromide … is recycled over halogen-rich sea ice surfaces" – these are the sources; recycling is different. Please clarify what you mean.

Line 187 – give the phases of the reactants and products in reaction 1

Line 191 – add reference to Simpson et al. GRL (32), 2005

Lines 198 to 200 should be used as the figure captions in Figs 4 to 7 – what is currently used is inadequate.

Line 205 typo: "maximum values in during late"

Line 209 and Figure 9a – how was sea ice area calculated; did it rely on the threshold of >15% sea ice in a pixel? More details are needed. But further, and importantly, there is no justification given of why the 130 to 190° sector is chosen. This is why proper assessment of air mass origin and history (as detailed above) becomes important.

Line 215 – explain why you used the data from Syowa and why they are the right data to use (presumably because they are similar latitude, but you need to say that)

Line 216 – Figure 9b relies entirely on correctly dating the cores, and correctly attributing the months to the measurements. This goes back to my concern about dating, above. There needs to be complete assurance that this has been done correctly.

Line 237 and Fig 10 – the figure shown originates in a paper by Anja Schoenhardt – the original reference needs to be provided, not the secondary one (Spoloar et al). Also, these data are not tropospheric measurements – they are vertical columns from space that have not been adjusted for any stratospheric component.

Line 244 – the polar night does not "start" in winter – please re-phrase

Section 4 – the Conclusions need to be re-visited in light of the above comments. In particular, the "Uniform satellite values of BrO and IO over Victoria Land confirm the snow measurements" is far too strong. They might "be consistent" with the snow measurements, but they do not confirm them. Finally, line 291, the halogens are not yet "proxies" as they are not rigorously demonstrated – they are potential-proxies, but not yet proven.

Table 3 – include also the median, given the statement in the text about high episodes, the median then becomes important.

Fig 1 – it would be useful to have latitude/longitudes on the lower maps or the inset maps to clarify how the zoomed-in map relates to the continent-wide ones.

---

## Referee Comment (RC3) · Anonymous Referee #3 · 11 Dec 2016

**Review of Bromine, iodine and sodium in surface snow along the 2013 Talos Dome – GV7 traverse (Northern Victoria Land, East Antarctica)**

This paper scrutinizes the seasonal cycles of sodium, bromine, and iodine in 7 shallow cores (2 m long spanning the 2010-2014 years) drilled in East Antarctica. The data are discussed with respect to sea-ice halogen sources. Data on bromine and iodine in snow and ice are very welcome since they are still rather rare and are potentially interesting for a better understanding of the halogen chemistry at high southern latitudes in the past, for instance.

However, major revisions of the manuscript are needed before I can recommend publication. The first major problem is that there are too many statements in the text that are not correct or oversimplified. Also the problem is that the manuscript totally ignores several relevant atmospheric studies conducted in Antarctica. Finally, in the discussion of data (figures 4 to 7) it would be nice to show not only the EF values but also the excess bromine relative to sodium with respect to seawater (or sea-salt aerosol) composition (see below).

**Line 38-51:** You should better explain to the readers how the examination of the bromine enrichment possibly helps to reconstruct sea-ice extent? Several atmospheric studies showed that not only sea-ice related processes (that are still not fully understood) but also open-ocean sea-salt emissions are important for the bromine chemistry (please cite Sander et al., 2003; Yang et al., 2005). Also relevant to your work is the recent atmospheric study conducted (JGR, 2016) at the coast and inland in the same East Antarctic region. In this study, both surface data and satellite observations indicate that in this region whereas the bromine chemistry is indeed maximum in spring, there is only a factor of two differences or less between spring and summer (suggesting again the importance of open ocean emissions in summer). This JGR paper also clearly showed that gaseous bromine species (that are water soluble and will be trapped in snow) largely dominate bromine aerosol. These gaseous species are likely responsible for the observed bromine enrichment in snow. Their atmospheric lifetime is far longer than the aerosol one due to a fast recycling on various surfaces (aerosol, snow grain). Therefore I have difficulty to understand the relationship between bromine enrichment in snow and the sea-ice extent ? For instance, whereas you mentioned in line 64 the noise introduced by transport in using sodium to reconstruct sea-ice, you have also to mention that since the bromine enrichment is related to gaseous species, the non irreversible trapping of the bromine species in snow would strongly handicap their use as proxy of sea-ice: please comment and cite Thomas et al. (2011).

Sander, R., et al. (2003), Inorganic bromine in the marine boundary layer: A critical review, Atmos. Chem. Phys., 3, 1301–1336, doi:10.5194/acp-3-1301-2003.

Yang, X., R. A. Cox, N. J. Warwick, J. A. Pyle, G. D. Carver, F. M. O'Connor, and N. H. Savage (2005), Tropospheric bromine chemistry and its impacts on ozone: A model study, J. Geophys. Res., 110, D23311, doi:10.1029/2005JD006244.

Legrand, M., X. Yang, S. Preunkert, and N. Theys (2016), Year-round records of sea salt, gaseous, and particulate inorganic bromine in the atmospheric boundary layer at coastal (Dumont d'Urville) and central (Concordia) East Antarctic sites, J. Geophys. Res. Atmos., 121, doi:10.1002/ 2015JD024066.

Thomas, J. L., J. Stutz, B. Lefer, L. G. Huey, K. Toyota, J. E. Dibb, and R. von Glasow (2011), Modeling chemistry in and above snow at Summit, Greenland-Part 1: Model description and results, Atmos. Chem. Phys., 11, 4899–4914, doi:10.5194/acp-11-4899-2011.

**Line 57 :** You should cite here the atmospheric study conducted by Grilli et al. (2013) that showed a less active iodine chemistry in East Antarctica compared to the case of west Antarctica (Saiz-Lopez et al., 2007). Such a difference should enhance the motivation to examine iodine in snow throughout Antarctica (west and East).

Grilli, R., M. Legrand, A. Kukui, G. Méjean, S. Preunkert, and D. Romanini, First investigations of IO, BrO, and $NO_2$ summer atmospheric levels at a coastal East Antarctic site using mode-locked cavity enhanced absorption spectroscopy, *Geophys. Res. Lett.*, 40, 1-6, doi:10.1002/grl.50154, 2013.

**Line 61-67 :** As far as I know, the pioneering finding of a correlation between sea-ice and MSA in snow from Curran et al. (2003) was never clearly confirmed by more recent snow studies. Furthermore, several atmospheric studies reported no evidence of such a link at the decadal scale (Weller et al., 2011; Preunkert et al., 2007).

Weller, R., D. Wagenbach, M. Legrand, C. Elsässer, X. Tian-Kunze, and G. König-Langlo (2011), Continuous 25-years aerosol records at coastal Antarctica: Part 1. Inter-annual variability of ionic compounds and links to climate indices, Tellus, Ser. B, 63, 901–919, doi:10.1111/ j.1600-0889.2011.00542.x.

Preunkert, S., M. Legrand, B. Jourdain, C. Moulin, S. Belviso, N. Kasamatsu, M. Fukuchi, and T. Hirawake, Interannual variability of dimethylsulfide in air and seawater and its atmospheric oxidation by-products (methanesulfonate and sulfate) at Dumont d'Urville (Coastal Antarctica) (1999-2003), *J. Geophys. Res.*, 112, doi:10.1029/2006JD007585, 2007.

**Line 68-71 :** Why do you introduce a discussion on post-depositional effect here ? The existence of post-depositional effect would have no effect on decadal or centennial scales.
Why "in particular Greenland" ? Please see and cite the recent work from Olivia Maselli (special issue in CP)

Maselli, O. J., Chellman, N. J., Grieman, M., Layman, L., McConnell, J. R., Pasteris, D., Rhodes, R. H., Saltzman, E., and Sigl, M.: Sea ice and pollution-modulated changes in Greenland ice core methanesulfonate and bromine, Clim. Past Discuss., doi:10.5194/cp-2016-49, accepted, 2016.

**Line 179 :** This number surprises me : 80% ? To what are related the missed (non sea-salt) sodium source that you consider to account for up to 20 % ? I don't think that the crustal source is large enough (at least for present-day climate), see and cite Weller et al. (2008).

Weller et al., Seasonal variability of crustal and marine trace elements in the aerosol at Neumayer station, Antarctica, Tellus 60B, 742-752, 2008.

**Line 201-204**, please also cite the atmospheric study from Wagenbach et al. (1998)

Wagenbach, D., F. Ducroz, R. Mulvaney, L. Keck, A. Minikin, M. Legrand, J. S. Hall, and E. W. Wolff (1998), Sea-salt aerosol in coastal Antarctic regions, J. Geophys. Res., 103, 0,961–0,974, doi:10.1029/97JD01804.

**Section 3.2:** Please report excess bromine either as Br - 6.2 $10^{-3}$*Na (or with error calculation Br – 8 $10^{-3}$*Na in winter and use brome depletion factor in aerosol taken from Sander et al. 2003 or Legrand et al. 2016 for spring and summer).

**Line 288-292:** Not sure that this is right: at least for bromine the homogeneity is also related to the atmospheric lifetime.

**End of the review**

---

## Author Comment (AC1) · 10 Jan 2017

We would like to thank Referee 1 for the review and for the helpful comments that motivate the research on this very current topic. We have responded to the comments and made the requested modifications. Particularly, the link with FYSI has been removed from the abstract and conclusion, and the transport processes issue influencing bromine and sodium (and bromine enrichment) deposition has been addressed (last figure). We acknowledge that an error was found in calculating the sea ice area, this have now been corrected.

It is well-established that the sea ice zone is a hotspot for Br (and with less certainty, I) chemistry. This has been shown by the high concentrations of BrO observed from satellite data in spring, and from ground-based data. The mechanism is understood to be that inorganic Br is activated from salty material (sea ice or aerosol). With this in mind, Spolaor et al (2013b in the present manuscript) found that Br was enriched in interglacial ice and depleted in glacial ice at Talos Dome. They proposed that this resulted from the halogen chemistry over sea ice, and that the enrichment or depletion might be used as an index of sea ice extent in the past. Their mechanism relied on their suggestion that Br/Na would reduce with distance from the sea ice edge, because the enriched material (as gas phase HBr) was deposited faster than the depleted (NaBr) sea salt aerosol. This was a surprising suggestion because previous work (Simpson et al, 2005) had suggested exactly the opposite (for the Arctic): that inland snow would be enhanced due to a longer lifetime of HBr. Clearly it is impossible to consider Br as a sea ice proxy until at least a reasonable understanding of the mechanism leading to temporal changes is understood, so studies attempting to deludicate this are very welcome. The present paper is aimed at doing this, by making a spatial transect (including seasonal information) of Na, Br and I in a part of East Antarctica. Unfortunately the location of the traverse is particularly badly chosen for such a study, because the sites sit between two marine areas, the Ross Sea and the main Southern Ocean. This has the effect that sites that are further from one potential source are nearer to the other, so that even if the data showed very clear trends, opposing interpretations would have been possible. As it happens there is little clarity in the data, which is not entirely the authors' fault but does mean that this is a paper which advances knowledge only incrementally. It is probably justified to consider publishing it after significant changes have been made, if only to indicate the complexity of the problem, and to show how premature it is to consider Br as a sea ice proxy until far more detailed and well-designed experiments and sampling campaigns have been carried out.
The paper itself is relatively short but with a very high number of tables and figures that really don't add to it, so among other things I would recommend losing some of these in the next version. There are also some unjustified interpretations (such as that in the abstract regarding Fig 9), which definitely must be modified.

Comments:
Abstract line 27-28. The last sentence is not justified. This is based on Fig 9c, with 4 data points. No statistics are given but I see no correlation at all, and a rough attempt to plot the data gave an r^2 very close to zero, utterly insignificant.
The sentence has been removed.

Sections 1 and 2 are generally OK, with two minor comments:
Line 36, remove the word layer. There is a "layer" of ozone in the stratosphere but there is no layer in the troposphere.
The term "layer" has been removed.

Line 68. The Rothlisberger article in a newsletter is not a good reference here. Better would be reference 3 or one of the other papers by Abram.

Röthlisberger reference has been replaced by Abram et al., 2013.

Line 162-167. While I agree that, taking old and new data into account, there is a tendency (as one would expect) of lower accumulation rates as we move further from the ocean, obviously this is somewhat undermined by the high value of 185 kg mˆ-2 aˆ-1 at TD. In the table the authors try to mitigate this by putting asterisks on "uncertain years", saying that their value is uncertain because the isotope signal is less clear. I don't think Fig 2 really justifies this – the least clear assignment at TD (2012) is at least as obvious as the one for 2011 at site 8 for example, and yet this has no asterisk against it in Table 2. I think a better way to handle this would be to remove the asterisks from the table, but to say that the inconsistency between the accumulation rate derived from the core at TD and that derived from the stake farm and previous measurements suggests that the isotopic assignments of years may be incorrect at TD, and that the profile contains more years than have been assigned.

We agree with the Referee comment with respect to Talos Dome. Therefore, the TD core is not used any further in the flux calculations nor in the %Brenr deposition. Only the profile is shown (on a depth scale) in the supplementary. The asterisks have been removed.

The sentence has been rephrased accordingly:

"The inconsistency between the accumulation rates derived from the core and those derived from the stake farm and previous measurements suggests that the isotopic assignments of years may be incorrect at this site, and that the profile contains more years than have been assigned. This core therefore is not used in further calculations. The fluxes of deposition of sodium, bromine and iodine in the other cores along the transect are calculated using the accumulation rates from this work."

Table 3 is unnecessary. Most of the information is anyway given in the text, but anyway a table like this that mixes different sites has no value that I can see. The table should be removed.

The table has been removed.

Line 189. This sentence is not really correct. Either enrichment or depletion can indicate that the reactions, believed to be focussed on sea ice, have taken place (not just enrichment). In fact the reaction (1) as shown leads only to a depletion of Br from the sea ice. It is only if the Br2 is eventually converted back to HBr that enrichment can occur, if more of this end product gas phase HBr is deposited nullifying the depletion in the aerosol phase. In addition, there are other ways to get such enrichment, as we know from the case of Cl, which is enriched or depleted compared to sodium due simply to production of HCl from the reaction between sulfuric or nitric acid with sea salt (e.g. H2SO4+2NaCl-> Na2SO4 + 2 HCl). A sentence that would be defendable would be "Therefore sea ice presence should lead to Br enrichment (after conversion of activated gas phase Br back to HBr) or depletion, depending whether deposition is dominated by the depleted sea salt aerosol or by the enriched gas phase HBr."

The sentence has been rephrased according to the Referee suggestion, with which we agree. The $Br(g) + HO_2(aq) \rightarrow HBr(g) + O_2$ reaction has been added.

Fig. 3 is OK, but I would have found it more useful if you had shown the distribution for individual sites or groups of sites. As a suggestion, you could show one distribution for sites GV7/8/7/6 (less than 100 km from ocean) and another distribution for the other sites.

Fig. 3 has been modified according to the Reviewer suggestion. The new figure now shows the two distributions of (TD, 10, 9) and (GV7, 8, 7, 6).

Fig 4, TD plot is obviously a problem, since you seem to believe that (Table 2) the years may be misassigned. This undermines your interpretations for this site. I'm afraid you can't have it both ways – either the TD accumulation is really high in which case the statements made about how accumulation varies with distance are not supported, or it is not in which case the year assignments shown in the TD section of this figure are wrong.

We agree on the Reviewer on this point. The Talos Dome core is not used in the following calculations (%Brenr and fluxes) and the variability figure (Fig. 4) has been removed. It is still shown in the supplementary material but plotted on a depth scale.

Fig 8 is really confusing because the map is upside down compared to that used in Fig. 1. In any case a similar figure is shown in Fig 1b. therefore please remove Fig 8, simply incorporating the additional information (basically the red box showing the 130-190E band) into Fig 1b.

We agree with the Referee on the comment. Fig. 8 has been removed and incorporated in Fig1 panel b, which now shows the sea ice concentrations (max and min) and the considered longitude sector. Note that the 'left' longitude sector has been rewritten as 170°W instead of 190°E. Some lat/long values have also been added to both panels as suggested by Referee2.

Line 215 – why is insolation from a site on the opposite side of Antarctica shown here? It is not even at the latitude of the sites here. Furthermore total solar radiation is very unlikely to be relevant, rather it is the radiation at the UV or near-UV wavelengths that might promote the relevant photochemistry. I suggest using a code such as TUV to calculate available radiation at relevant wavelengths.

We agree with the Referee comment and Fig9a+b have been modified accordingly (now Fig 7). The daily average total downwelling spectral irradiance has been calculated using Tropospheric Ultraviolet and Visible Radiation Model (TUV), as suggested, in the [300,500] nm interval. The choice on the wavelength interval was made based on [Saiz-Lopez et al.: Measurements and modelling of I2, IO, OIO, BrO and NO3 in the mid-latitude marine boundary layer, (2006), fig 12]. The daily averages values were calculated for year 2012 (15th day, representative for the month). The model calculations were set at 71° S, 158° E. The normalized trend (Fig 7b, magenta) is very similar and thus the interpretation of combined effects of sea ice and radiation triggering photochemistry.

Line 205, and Fig 9b. It's hard from Figs 4-7 to really see the seasonality, which you describe as max in late spring/summer for Br. Since you must have calculated it to get to Fig 9b, please add a plot of this sort for each site so we can see how consistent the seasonality is at different sites. This could be shown as individual lines underlying the error band in Fig 9b for example, or as a separate panel if this is less confusing.

The seasonality at the different sites are now shown in Fig. 7 (having excluded Talos Dome and year 2013 in core 10, consistent with the previous comments). The blue band shows 1sigma variability of the different cores.

Line 221-3. The comparison of the seasonality with the product of radiation and sea ice extent is interesting but it does not "demonstrate" the dependency of Br enrichment on their combined effect, rather it is "consistent with" the idea that there is such a dependency. Please adjust the text.

The sentence has been rephrased as: "Such comparison suggests that the combined effect of sea ice and insolation drives the seasonality of bromine enrichment."

Lines 225-7. As in the abstract, the correct characterisation of Fig 9c is that, with only 4 years of data, no relationship can be discerned. It is not scientific to say that there is a relationship for 3 years and not for the fourth – statistically this means there is no relationship. You just have to admit that there are not enough data here to know whether Br_enr can be used as an indicator of FYSI. In fact (and related to the next comment) nothing you have written here says why you would expect the enrichment to be related to the FYSI area; what mechanism are you envisaging? If, as in the earlier Spolaor paper, it is via distance from the ice edge (through whatever process) then this should be somewhat reflected in the difference between sites, hence my next comment.

The sentence has been removed as in the abstract/conclusion, but the plot has been left. In the introduction, we now better explained the mechanism linking bromine enrichment to FYSI area. The idea is

that more first year sea ice would lead to bromine enrichment over its surfaces. Depletion can be observed at the deposition site if depleted aerosol dominates gas phase HBr or if HBr is deposited more rapidly.
At glacial/interglacial timescales, the situation is much different than looking at present conditions, since the source (FYSI) is much further away in the glacial (on the basis of more extensive MYSI zone). The interpretation of a glacial/interglacial time serie than relies on a combined effect of source (FYSI/MYSI) and transport processes. The paper Spolaor et al. Canadian Arctic sea ice reconstructed from bromine in the Greenland NEEM ice core, Sci. Rep., 6, doi:10.1038/srep33925, 2016b addresses the topic at such timescales. Unfortunately, it was not possible to test transport mechanisms with this traverse data, since as we move further from one source we approach the other one (see next comment).
The traverse from MZS to Dome C will surely provide more clues on this topic.

Section 3.3 and Fig. 11. I agree that the Br pattern looks similar to the Na pattern. But this then begs the question, what is the pattern of Br_enr. My impression from this figure and the data in Figs 4-7 is that Br enr is probably rather flat with distance. Since this was the issue that Spolaor suggested controlled the TD glacial-interglacial change, it deserves to be shown and discussed in that context. Please add a discussion.
Figure 11 has been modified (now Fig. 9) and now shows the spatial pattern of Brenr in the second column, and bromine and sodium in the first one. The paragraph has been extended to discuss the point in question:
"The pattern of bromine enrichment is linked among other things to the different bromine fractionations during the transport in the gas phase and the aerosol phase, compared to sodium. Unlike sodium and bromine, no decrease is observed for bromine enrichment from our data (Fig. 9, second column), although no clear trend can be inferred. This can be due to the multiple origins of air advection (Ross sea /Indian ocean), to the uneven strength of source areas or because the distances are not large enough for any difference to be reliably observed."

**Below the new figures:**

**Figure 1.** (a) Schematic map of the traverse area and coring sites, marked with stars. The cores were drilled between Nov 20th 2013 and Jan 8th 2014 (early austral summer). (b) Maximum (left, August 2011) and minimum (right, January 2010) sea ice concentrations in the 130°E-170°W sector for the 2010-2013 time interval covered by the core records (NSIDC data from Meier et al., 2013). The traverse location is marked with an ellipse.

[Figure]

**Figure 3**. Distribution of bromine enrichment values within cores TD, 10, 9 (blue) and GV7, 8, 7, 6 (red). The dashed line indicates the seawater value ($Br_{enr} = 1$).

[Figure]

**Figure 4.** Variability of δ¹⁸O (upper panel), Na (middle top panel, left axis), Br (middle top panel, right axis), Brenr (middle bottom panel, left axis), nssBr (middle bottom panel, right axis), and I (bottom panel) in cores 10 (left) and 9 (right). Thick lines represent 3-month running means of the raw data (circles).

[Figure]

**Figure 5.** Variability of δ¹⁸O (upper panel), Na (middle top panel, left axis), Br (middle top panel, right axis), Brenr (middle bottom panel, left axis), nssBr (middle bottom panel, right axis), and I (bottom panel) in cores GV7 (left) and 8 (right). Thick lines represent 3-month running means of the raw data (circles).

[Figure]

**Figure 6.** Variability of δ¹⁸O (upper panel), Na (middle top panel, left axis), Br (middle top panel, right axis), Brenr (middle bottom panel, left axis), nssBr (middle bottom panel, right axis), and I (bottom panel) in cores 7 (left) and 6 (right). Thick lines represent 3-month running means of the raw data (circles).

[Figure]

**Figure 7. (a)** Monthly values of sea ice area (blue) within the 130°E-170°W sector from 2010 to 2013 (±1σ, month variability) and daily average (24 hours) total downwelling spectral irradiance (red), calculated using the TUV model at 71° S, 158° E. Each irradiance calculation was set the 15th day of each month, in 2012. **(b)** Seasonality of annual bromine enrichment along the traverse: the monthly trend shows a seasonal feature with maximum in Spring. Each line refers to a core of the transect (±1σ, shaded blue area). The month averages are displayed in black. The systematic uncertainties associated to the dating are shown as verticals error bars. The magenta band represents the product distribution of normalized sea ice area and insolation, expressed in annual percentage. **(c)** Monthly sea ice area values (blue) from 2010 to 2013, with annual values of FYSI (red) and averaged bromine enrichment (black).

[Figure]

**Figure 9.** Mean annual fluxes of sodium (blue, left axes), bromine (red, right axes), iodine (green) and bromine enrichment values (black), as a function of distance from the Indian Ocean. Each dot represents a location along the traverse.

[Figure]

distance from Indian Ocean (km)

**Supplementary material**

**The following figure shows the measurements related to the Talos Dome core.**

**Figure.** Variability of $\delta^{18}O$ (upper panel), Na (middle top panel, left axis), Br (middle top panel, right axis), Brenr (middle bottom panel, left axis), nssBr (middle bottom panel, right axis), and I (bottom panel) in the Talos Dome core on a depth scale. Thick lines represent 15 cm running means of the raw data (circles).

[Figure]

---

## Author Comment (AC2) · 10 Jan 2017

**Author comment on**

**"Bromine, iodine and sodium in surface snow along the 2013 Talos Dome – GV7 traverse (Northern Victoria Land, East Antarctica)" by Maffezzoli et al.**

**Anonymous Referee #2**

We would like to thank Referee 2 for the extensive review and the time taken to read the paper. The introduction has been extensively revised, especially for the analytical and dating parts. We acknowledge that an error was found in calculating the sea ice area, this has now been corrected.

**Major concerns**

**Analytical information** is lacking. The authors present very few analytical details, instead refering readers to the Spolaor et al, 2013a paper. However, some information will be specific to the Maffezzoli paper and should be included, and some additional information would be useful.
We agree with the reviewer comment and have added the requested information. The citation of the method was wrong (2013a). The correct reference was the 2014 paper. The analytical part has been corrected and expanded including the following details (see below).
For example, what was the residual standard deviation for Br, I, and Na for the current work?
"The reproducibility of the measurements was carried out by repeated measurements of standard samples within the calibration range. The residual standard deviations were respectively 5 % (bromine), 3 % (sodium) and 2 % (iodine). "
How were the standards prepared – gravimetric or volumetric methods?
The standards were prepared by gravimetric method.
Standard concentrations ranged between 10 and 4000 ppt, but for for which species? Spoloar 2013 refers to iodine and bromine being calibrated in this way – but what standards were used for sodium? Presumably halogen standards were separate from sodium standards – did this introduce any uncertainties into the analyses?
We thank to referee for the comment.
The standard solutions were prepared from separate stock 1000 ppm standard solutions of the three analytes. The primary solution was then diluted for the calibration purposes into 6 bromine and iodine standards (0.01, 0.05, 0.1, 0.5, 1 and 4 ppb) and 6 sodium standards (0.5, 1, 5, 10, 50 and 100 ppb). Previous tests have suggested that mixing of standards didn't change the response signal compared to separate standard solution.
I am not so familiar with this technique – does it analyse Na, I, and Br in a single run? If not, what uncertainties does this introduce in terms of instrument drift etc? Does the method use columns to separate out the elements of interest? If so, what columns were used?
Yes, the analytes are detected almost simultaneously (instrument detects each analyte 5 times and averages the response). No column separation was used: the sample was introduced with a cyclonic Peltier-cooled spray chamber (ESI, Omaha, USA) directly injected into the ICP-SFMS at a flow rate of 0.4 mL min$^{-1}$.
Was any reference material used to really pin down the analytical technique, given the extremely low concentrations being measured?
There is no halogen certified reference material that we know of.
Exposure of samples to direct light was minimised, but how long were they actually exposed for?
"The samples were exposed to laboratory light for a maximum time of one hour."
How long were the samples melted for before they were analysed?
"The samples were melted maximum one hour before measurements."
Were there any repeat analyses of samples carried out?
Every data point is the average of 5 almost simultaneous instrumental detections. Precautions were taken to assure that each of the 5 detections were consistent (less than 2% variations).
How much liquid water was needed for each analysis?
"The sample flow was 0.4 mL min$^{-1}$ for a total sample volume of approximately 5.0 mL".

Presumably the samples were re-frozen before being shipped to Copenhagen for isotopic analysis. Is this likely to have introduced any problems/uncertainties into the analyses?

Sub sample aliquots for isotopic measurements were taken during the melting phases. The aliquots were immediately refrozen and shipped frozen to Copenhagen. We don't think that this procedure introduced any major uncertainty since the time span in which the samples were melted was on the order of 10 minutes and precautions were taken to minimize evaporation (septum-sealed glass vials were used to these aliquots).

Finally, the manuscript states that during sampling, every tool was repeatedly cleaned with ultrapure water – how many times is "repeatedly". What effort was made to ensure that the tools were clean?

Rephrased as: "Every tool was cleaned each time a piece of sample was decontaminated into three serial baths of ultrapure water, which was changed every 10 washes."

**The analytical section has been reviewed:**

**2.2 Analytical measurements**

Total sodium (Na), bromine (Br) and iodine (I) concentrations were determined by Inductively Coupled Plasma - Sector Field Mass Spectrometry (ICP-SFMS Element2, ThermoFischer, Bremen, Germany) at Cà Foscari University of Venice, following the methodology described in Spolaor et al., 2014.

The samples were melted one hour before measurements. During this time exposure from direct light was reduced by covering them with aluminum foils, minimizing bromine and iodine photolysis reactions.

The introduction system consisted of a cyclonic Peltier-cooled spray chamber (ESI, Omaha, USA). The operational flow rate was kept at 0.4 mL min$^{-1}$, for an overall sample volume of 5.0 mL. Each sample determination consisted of 5 instrumental detections (less than 2% variations between them). The 5 values were then averaged to provide the final quantification.

Each analytical run (10 samples) ended with a HNO$_3$ (5%) and UPW cleaning session of 3 min to ensure a stable background level throughout the analysis.

The external standards that were used to calibrate the analytes were prepared by gravimetric method by diluting separate stock 1000 ppm IC solution (TraceCERT® purity grade, Sigma-Aldrich, MO, USA) of the three analytes into a primary solution, which was further diluted for into 6 bromine and iodine standards (0.01, 0.05, 0.1, 0.5, 1 and 4 ppb) and 6 sodium standards (0.5, 1, 5, 10, 50 and 100 ppb).

The calibration regression lines showed correlation coefficients $R^2>0.99$ (N=6, p=0.05). The detection limits, calculated as three times the standard deviation of the blanks, were 50 and 5 ppt for bromine and iodine respectively and 0.8 ppb for sodium. The reproducibility of the measurements was carried out by repeated measurements of standard samples within the calibration range. The residual standard deviations (RSD) were respectively 5 % (bromine), 3 % (sodium) and 2 % (iodine).

Procedural UPW blanks were analyzed periodically to test the cleanliness of the instrument lines.

Stable isotopes of water ($^{18}$O and D) measurements were conducted on sub sample aliquots, which were immediately refrozen and shipped to the Center for Ice and Climate (Copenhagen, Denmark). Analyses were carried out using a Cavity Ring-Down Spectrometer (Picarro, Santa Clara, USA) using the method described by Gkinis et al. (2010). Septum-sealed glass vials were used for these measurements to prevent any sample evaporation during the experimental phases.

**Dating**: The conclusions drawn in the paper rely heavily on correct dating of the cores, i.e. correctly allocating samples to specific seasons. This is done using stable water isotopes, and the manuscript states (line 140) that "isotope ratio minima (representing mid-winter) can be easily identified". But the authors must give the criteria used for such identification, if other than just by eye. As the authors state, there is clearly a question over assignment of the 2013 mid-winter – how did the authors select the one chosen? I also wonder what happened to the top layer of core 6..? Was it damaged during sampling? The accumulation rate of core 10 is less than core 9, so one would expect the snow at 2m depth to be older in core 10 than in core 9; however, this is not the case as presented in the paper.

It makes me wonder whether mid-winter assignments in core 10 are out by a year? Certainly, more information on the criteria used to allocate winter minima must be given for cases where it is not completely clear.

If the Referee refers to assignment of year 2013 in core 10, it is clear that the surface of core 10 is loaded with a layer of snow with mixed isotopic value. The effect of this layer of disturbed snow is to push the ordered stratigraphy downwards, resulting in 'younger' snow at the bottom compared to core 9. Therefore, winter 2013 in core 10 is suggested, but summer values (beginning of 2013) are clearly visible (and that is a tie point in Fig. 2).

Year 2013 in core 10 has been removed from the calculation of % annual bromine enrichment (now Fig. 7b) and in the calculation of the fluxes (now Fig. 9) because of the these reasons.

The top part of core 6 has probably been lost due to wind erosion. Though, we concluded from the isotopic values that the surface is the start of a 'cold' period.

The methodology to assign winters has been written more clearly. Lines 139-154 have been rephrased:

"The cores were dated based on the seasonal variations identified in the stable water isotopes (both $\delta^{18}O$ and $\delta D$). Midwinters were associated to the relative minima of the isotopic curves (Fig. 2). In case a winter isotopic plateau was found, the center of the plateau was associated to midwinter depth (2011 in core GV7; 2012 and 2011 in core 8; 2010 in core 6). Almost all the cores cover the period between 2010 and late 2013, providing four years of snow deposition. The only exception is represented by core 6, whose upper layer is missing. The annual deposition signal looks less clear in the two cores that were drilled at the sites with the highest elevation and the closest to the Ross Sea, cores TD and 10, and especially for 2013 in core 10. The two sites are probably the most affected by surface remobilization and isotopic diffusion due to low accumulation. Indeed, non-uniformities in the shallow snow layers such as sastrugi, dunes, wind crusts and other features have been identified as an important aspect of the surface morphology around the Talos Dome area (Frezzotti et al., 2004; 2007).

The annual accumulation rates were calculated by selecting the depth intervals included within consecutive maximum or minimum $\delta^{18}O$ values (Table 2). Each snow layer within this interval (i.e. sampling resolution, 5 cm) was multiplied by the density of the snow at that depth, the density curves having the same resolution. The contributions were summed over the annual thickness. Table 2 also includes accumulation rates in Victoria Land reported from previous studies. The GV5 site is located between sites 10 and 9 (Fig. 1)."

Please explain fully how accumulation rates were calculated, rather than just saying (line 153) that measured density profiles were accounted for – explain the method.

The methodology to calculate the accumulation rate has been written more clearly, see comment above.

Was only 1 core taken in each site? How do you account for variability between cores drilled at the same site? If this is not considered important, then please say so and explain why.

The main goal of the traverse was to compare halogen seasonality of deposition in a wide area, since no transect before this one has ever been performed for this purpose. It would have been surely interesting to investigate the spatial variability in each location but this was not possible due to logistic constrains. The traverse between Mario Zucchelli and Concordia stations (planned for 2017/18) will provide a more complete picture especially to elucideate gas-phase vs aerosol fractionation and deposition.

The authors must provide justification on why they chose to use the stake farm data for Talos Dome over their own method. It is not enough to reject your method, which is used on all other sample sites, just because it does not agree with stake farm data at TD. This selection raises concerns over the validity of the method used for all the other sites.

Talos Dome, is known to be a difficult location due isotopic diffusion and non-uniformities (Frezzotti et al., 2004; 2007). Therefore, and because the value in Talos Dome is the only one that is inconsistent with literature values, we decided to disregard the core from the % annual bromine enrichment (now Fig. 7b) and the flux calculations (now Fig. 9) and show its chemistry variability in the supplementary on a depth scale. The other values are consistent with published data, therefore we used them.

**Below are more minor comments:**
- Line 16: halogen chemistry does not only occur through release of sea salt rich aerosols; various saline condensed phases have been suggested;
Rephrased: " .. sea salt aerosols and other saline condensed phases ..".

- line 18: the statement "halogen species in polar snow samples are shown to be closely related to sea ice extent" is too strong – there is clearly a link to sea ice, but this is far from quantified, so how close it is related is not yet known.
'closely' has been removed.

-line 25: the transect revealed homogeneous fluxes" – what type of fluxes? Air-to-snow..? Snow-to-air..?
Modified to 'air-to-snow'.

-line 27: "flux measurements are consistent with the uniform values of BrO and IO"... uniform in time or space..?
Uniform in space.
Rephrased: " .. BrO and IO concentrations detected from satellites over the traverse area."

Line 36 – there is no tropospheric ozone layer; remove the word "layer"
The word ''layer'' has been removed.

Line 36 – there are many other papers that should be referenced here other than Barrie et al. There are also some excellent reviews, e.g. Simpson et al. 2007, Abbatt et al. 2012, that should also be included here.
The references have been added.

Line 38 – explain why young sea ice surfaces have high salinity, and how they are thus a source of bromine compounds (presumably you mean to the atmosphere). Also, what are "bromine halides"..??
This sentence has been reviewed:
"Although the ocean is the main reservoir of sea salts, various condensed phases of high salinity are found on young sea ice surfaces. During seawater freezing, brine is separated from the frozen water matrix and expulsion processes lead to both upward and downward movement, as temperature decreases (Abbatt et al., 2012). Therefore, high salinity brine, frost flowers and salty blowing snow make newly formed sea ice surfaces a highly efficient substrate for inorganic halogen halides and for their activation and release in the atmosphere (Saiz-Lopez et al., 2012b). "

Line 43 – include the reactions and improve the wording here
The sentence has been reviewed:
"Reactive halogen species are involved in cyclic reactions between halogen radicals, their oxides and ozone. Reactions R1-3 show the main reactions for bromine. Atomic bromine radicals result from photolysis of molecular bromine, leading to formation of bromine oxide, BrO, through the uptake of ozone:

$$Br_2 \rightarrow 2Br \qquad\qquad\qquad (1)$$

$$Br + O_3 \rightarrow BrO + O_2 \qquad\qquad (2)$$

$$BrO + BrO \rightarrow Br + Br + O_2 \qquad\qquad (3)$$

Self reaction of BrO may form 2 bromine atoms (85%) or a $Br_2$ molecule (15%) which is readily photolyzed. The mechanism has a catalytic behavior that destroys ozone."

Line 45- MAX-DOAS generally refers to a ground-based instrument technique; SCIAMACHY is a satellite-borne uv-vis-nir spectrometer that quantifies BrO and IO columns using the DOAS technique. Please re-word. Also state which satellite instrument you refer to.

The sentence has been reviewed:

"High concentrations of tropospheric vertical columns of BrO and IO have been confirmed by SCIAMACHY (SCanning Imaging Absorption spectroMeter for Atmo- spheric CartograpHY) satellite observations over Antarctic sea ice (Schönhardt et al., 2012)."

Line 47 – "from halogen-rich condensed phases (e.g. sea salt aerosol)" – there are likely to be others as well.

Rephrased as: "Bromine can then be recycled and re-emitted from halogen-rich condensed phases (such as sea salt aerosol or other saline solutions)"

Line 49: Such reactions ... lead to enhancement of bromine in the deposition in the surface snowpack - give a reference to support this statement.

We agree with the Referee, as this is a central point. The original hypothesis of bromine enrichment has been recently supported by a chemistry model in the 2016b paper. The reference is added and the sentence made clearer. The topic is further addressed later on in this paper, answering to Referee1 comment in Line 189.

"Such reactions, known as bromine explosions, lead to enhanced bromine in the atmosphere. A recent 1D chemistry model simulation predicted an increase of bromine deposition on surface snowpack after 24/48 hours of recycling over first year sea ice (Spolaor et al., 2016b)."

Line 51 – is the Spolaor reference 2016 a or b?

It's reference 2016 a and b, updated.

Line 52 – the Vogt paper is a modelling study, so does not include primary information on iodine sources

The Vogt reference has been replaced by the Saiz-Lopez et al., 2012a review.

Line 66 – DMS is produced from DMSP, not directly from phytoplankton

Rephrased as: "…dimethylsulfide (DMS), which is produced by phytoplankton synthesis of DMSP".

Line 70- I believe Mulvaney and Pasteur were the first to report post-depositional movement of MSA in ice cores.

The references have been added and "mostly in Greenland" has been removed.

Line 78 – "Back trajectory calculations show that favourable events of air mass advection from the sea ice surface to TD are rare but likely to occur" – what calculations are these? Which model? Did the authors run them, or are they in a published paper? If the latter, then give references. If the former, give more information. An example plot would be useful with some sort of analysis of air mass origin.

This is a central point in the choice of sea ice sector and in the interpretation of the fluxes, so we appreciate the importance of communicating this clearly.

An extensive study in this area on the provenance of air masses is presented in the paper by Scarchilli et al. (2010) which is mentioned in section 3.3. This paper shows that (analyzing daily back trajectories from 1980 to 2001) air masses arriving in Talos Dome are clustered into two main sources: the Ross sea and the Indian

ocean sector (Fig. 4 in the Scarchilli paper). Moving from Talos Dome to the Indian ocean (where cores GV7, 8, 7 and 6 were drilled) we expect this sector to be more and more important. Hence our choice of the sea ice sector.
Another paper, Sala et al. (2008) has studied provenance of air masses from back trajectory calculations from 1990 to 2002, pointing out sea ice influence at Talos Dome.
Both references are now mentioned here.

Actually the point being made in paragraph of line 72 to line 82 is not clear.
The paragraph has been rewritten to state more clearly that this area is influenced by marine and sea ice air advection. This has been demonstrated by measurements of chemical species in the snow (traverse studies and longer cores) and back trajectory calculations. The 'dust' part and its refences have been removed.

"Victoria Land has been intensively studied for the past two decades. The Taylor Dome (Grootes et al., 2001) and Talos Dome (Stenni et al., 2011) deep ice cores respectively provide 150 kyr and 300 kyr climatic records directly influenced by marine airmasses. Sala et al. (2008) pointed out the presence of marine compounds (ikaite) at Talos Dome, typically formed at the early stages of sea ice formation. Their back trajectory calculations also showed that favourable events for air mass advection from the sea ice surface to Talos Dome are rare but likely to occur. An extensive study by Scarchilli et al. (2011) on provenance of air masses has shown that Talos Dome receives 50% of its total precipitation from the west (Indian Ocean), 30% from the east (Ross Sea and Pacific Ocean) and approximately 15% from the interior. Within the framework of the ITASE program (International Trans-Antarctic Scientific Expedition, Mayewski et al., 2005), several traverses were carried out to evaluate the spatial patterns of isotopic values and chemical species linked to marine influence (Magand et al., 2004; Proposito et al., 2002; Becagli et al., 2004, 2005; Benassai et al., 2005)."

Line 90 and 93 – "Indian **ocean** sector"
Corrected.

Line 92 – Friess et al did not report Br in their snow pit.
Corrected.

Line 104 – it would help the reader to give the distance between the cores.
Table 1 has been updated with un extra column showing core-to-core distances in km.

Line 184 – The description of bromide release needs improving. e.g."bromide … is recycled over halogen-rich sea ice surfaces" – these are the sources; recycling is different. Please clarify what you mean.
Rephrased: "Despite bromine being a sea salt marker like sodium, it is activated when gas phase HOBr oxidizes bromide over halogen rich sea ice surfaces (i.e. first year sea ice, FYSI) and suspended sea salt aerosol, and exponentially released as $Br_2$".

Line 187 – give the phases of the reactants and products in reaction 1
The phases have been written. The following reaction, $Br(g) + HO_2(aq) \rightarrow HBr(g) + O_2$(g), has also been added in response to Referee1 to motivate the possibility of enrichment in the following sentence.

Line 191 – add reference to Simpson et al. GRL (32), 2005
Added.

Lines 198 to 200 should be used as the figure captions in Figs 4 to 7 – what is currently used is inadequate.
Captions have been rewritten accordingly.

Line 205 typo: "maximum values in during late"
Corrected into 'maximum values in late..'.

Line 209 and Figure 9a – how was sea ice area calculated; did it rely on the threshold of >15% sea ice in a pixel? More details are needed. But further, and importantly, there is no justification given of why the 130 to 190° sector is chosen. This is why proper assessment of air mass origin and history (as detailed above) becomes important.

The sea ice area was calculated by multiplying the sea ice concentration value in each pixel by the pixel area (25X25 km$^2$). A mistake in the sea ice computation was found: the new figure (Fig. 7) shows the correct values (the trend doesn't show any difference). No threshold was applied to the sea ice concentration values.

As in the Referee comment at line 78, we referred to Scarchilli et al. (2011) for the choice of the sea ice sector. The sector boundaries have been rewritten as 130°E-170°W throughout the text.

The sentence has been reviewed:
"Sea ice area in the 130°E-170°W sector was calculated for the 2010-2013 period using publicly available NSIDC passive microwave sea ice concentration data (Meier et al., 2013), by multiplying the sea concentration value in each grid pixel by the area of the pixel (25 x 25 km$^2$) and integrating over the domain. The longitude sector was decided on the basis of Scarchilli et al. (2011), who concluded that air masses arriving in this area originate from the Ross sea and from the Indian ocean sector, by analyzing 5 day back trajectories from 1980 to 2001."

Line 215 – explain why you used the data from Syowa and why they are the right data to use (presumably because they are similar latitude, but you need to say that).
In sight of a similar comment from Reviewer 1, we calculated the solar irradiance at 71°S, 158°E with Tropospheric Ultraviolet and Visible (TUV) model. The figure has been updated with such values (no visible difference appears in the normalized SI*radiation magenta band). The caption has also been revised.

Line 216 – Figure 9b relies entirely on correctly dating the cores, and correctly attributing the months to the measurements. This goes back to my concern about dating, above. There needs to be complete assurance that this has been done correctly.
Year 2013 of core 10 and the Talos Dome core have been removed this figure, in accordance to the previous comments and findings on dating and accumulation values. Each contribution of each core to the overall deposition trend has been added on top. The blue band shows 1sigma.

Line 237 and Fig 10 – the figure shown originates in a paper by Anja Schoenhardt – the original reference needs to be provided, not the secondary one (Spoloar et al). Also, these data are not tropospheric measurements – they are vertical columns from space that have not been adjusted for any stratospheric component.
The picture has been produced for the Spolaor et al. (2014) paper; the reference therefore has been left. "Tropospheric" has been removed.

Line 244 – the polar night does not "start" in winter – please re-phrase
Rephrased as: "... in winter, when absence of sunlight inhibits photoactivation."

Section 4 – the Conclusions need to be re-visited in light of the above comments.
The conclusions have been reviewed:
"The 2013/14 Talos Dome – GV7 traverse provided an opportunity to expand the existing sodium dataset in Victoria Land and investigate important features of bromine and iodine temporal and spatial variabilities, so far only available in Antarctica at Law Dome and Neumayer station.

The accumulation rates agree with previous studies, with increasing values from the Ross Sea to the Southern Ocean. Accumulation rates calculated for Talos Dome are higher than previously reported, likely caused by isotopic diffusion and remobilization at this site. Further studies are required at this site in order to access the

reproducibility of the climate signal. The locations near the Southern Ocean exhibit high variability due to the higher accumulation.

Sodium and bromine concentrations in the snow samples result in a positive bromine enrichment to seawater, confirming the sea ice influence in the area for the extra bromine deposition. While sodium does not capture clear sub-annual variations associated with sea ice, bromine enrichment shows consistent seasonal variabilities with late spring maxima. It is possible to relate such seasonality to the combined effect of sea ice growth and sunlight, which trigger photochemistry above fresh sea ice. The timing of deposition is coherent among Victoria Land, Law Dome (Indian sector) and Neumayer (Atlantic sector). Iodine shows an average value of 0.04 ppb, similar to background values observed in the Antarctic coastal locations of Law Dome and Neumayer. Unlike those locations, low iodine annual variability and no consistent seasonality of the signal are observed in the traverse samples.

The spatial variability study reveals homogeneous fluxes of Na, Br, and I over the transect length, with an increase in absolute values and variability at the sites close to the Indian Ocean, due to high accumulation and proximity to the coasts. Uniform satellite values of BrO and IO over Victoria Land are consistent with the snow measurements. A fractionation due to distance of these potential proxies is not found probably due to the combined double input of air masses from the Ross Sea and the Indian Ocean.
A transect covering larger distances and directed towards the interior of the plateau would give an insight on this feature, especially clarifying the spatial pattern of bromine enrichment with respect to differences in gas-phase and aerosol depositions. "

In particular, the "Uniform satellite values of BrO and IO over Victoria Land confirm the snow measurements" is far too strong. They might "be consistent" with the snow measurements, but they do not confirm them.
Rephrased accordingly.
Finally, line 291, the halogens are not yet "proxies" as they are not rigorously demonstrated – they are potential-proxies, but not yet proven.
"Potential" has been added.

Table 3 – include also the median, given the statement in the text about high episodes, the median then becomes important.
The table was removed as suggested by Referee #1.

Fig 1 – it would be useful to have latitude/longitudes on the lower maps or the inset maps to clarify how the zoomed-in map relates to the continent-wide ones.
The lower map has been modified as suggested by Referee #1 (and Figure 8 has been removed). It now shows the max/min sea ice concentrations and the sea ice sector which is considered. Some lat/long values in both top and bottom panels have been added for clarification.

**Added references**

[revised manuscript text omitted]

distance from Indian Ocean (km)

**Supplementary material**

**The following figure shows the measurements related to the Talos Dome core.**

**Figure.** Variability of δ¹⁸O (upper panel), Na (middle top panel, left axis), Br (middle top panel, right axis), Brenr (middle bottom panel, left axis), nssBr (middle bottom panel, right axis), and I (bottom panel) in the Talos Dome core on a depth scale. Thick lines represent 15 cm running means of the raw data (circles).

[Figure]

---

## Author Comment (AC3) · 10 Jan 2017

**Author comment on**

**Review of Bromine, iodine and sodium in surface snow along the 2013 Talos Dome – GV7 traverse (Northern Victoria Land, East Antarctica)**

**Anonymous Referee #3**

We would like to thank Referee 3 for the review and the constructive criticism to the points that have been questioned. We reply to them and made the requested modifications. We acknowledge that an error was found in calculating the sea ice area, this has now been corrected. The introduction has been revised and almost all the suggested references have been added. The nssBr (non-sea-salt bromine) variability is not shown in the plots.

This paper scrutinizes the seasonal cycles of sodium, bromine, and iodine in 7 shallow cores (2 m long spanning the 2010-2014 years) drilled in East Antarctica. The data are discussed with respect to sea-ice halogen sources. Data on bromine and iodine in snow and ice are very welcome since they are still rather rare and are potentially interesting for a better understanding of the halogen chemistry at high southern latitudes in the past, for instance. However, major revisions of the manuscript are needed before I can recommend publication. The first major problem is that there are too many statements in the text that are not correct or oversimplified. Also the problem is that the manuscript totally ignores reveral relevant atmospheric studies conducted in Antarctica. Finally, in the discussion of data (figures 4 to 7) it would be nice to show not only the EF values but also the excess bromine relative to sodium with respect to seawater (or sea-salt aerosol) composition (see below).

**Line 38-51:** You should better explain to the readers how the examination of the bromine enrichment possibly helps to reconstruct sea-ice extent? Several atmospheric studies showed that not only sea-ice related processes (that are still not fully understood) but also open-ocean sea-salt emissions are important for the bromine chemistry (please cite Sander et al., 2003; Yang et al., 2005).

We acknowledge the presence of open-ocean sources and the related model studies. The 'open-ocean emissions' are now added (along with the references) to the text, however we point out that sea ice is a necessary source for bromine recycling leading to the BrO concentrations observed from satellites and seasonality of bromine (and br enrichment) peaks in ice core records in springtime. The importance of sea ice for bromine recycling is demonstrated in the chemical transport model results presented in Spolaor et al., 2016b. In addition, a more recent paper by Yang et al., 2008 - Sea salt aerosol production and bromine release: Role of snow on sea ice, indicates "that the importance of blowing snow (on sea ice) can be more than an order of magnitude larger than sea salt aerosol production rates from the ocean under typical weather conditions". We therefore also included this reference (being the same author).

The part related to the use of bromine enrichment has been made more clear:
"Although the ocean is the main reservoir of sea salts, various condensed phases of high salinity are found on young sea ice surfaces. During seawater freezing, brine is separated from the frozen water matrix and expulsion processes lead to both upward and downward movement, as temperature decreases (Abbatt et al., 2012). Therefore, high salinity brine, frost flowers and salty blowing snow make newly formed sea ice surfaces a highly efficient substrate for inorganic halides and for their activation and release in the atmosphere (Saiz-Lopez et al., 2012b, Yang et al., 2008). Some studies have pointed out the role of open-water sea salts as a significant bromine source (Yang et al., 2005; Sander et al., 2003).

Reactive halogen species are involved in cyclic reactions between halogen radicals, their oxides and ozone. Reactions R1-3 show the main reactions for bromine. Atomic bromine radicals result from photolysis of molecular bromine, leading to formation of bromine oxide, BrO, through the uptake of ozone:

$$Br_2 \rightarrow 2Br \tag{1}$$

$$Br + O_3 \rightarrow BrO + O_2 \tag{2}$$

$$BrO + BrO \rightarrow Br + Br + O_2 \tag{3}$$

Self reaction of BrO may form 2 bromine atoms (85%) or a $Br_2$ molecule (15%) which is readily photolyzed. The mechanism has a catalytic behavior that destroys ozone.

High concentrations of tropospheric vertical columns of BrO and IO have been confirmed by SCIAMACHY (SCanning Imaging Absorption spectroMeter for Atmospheric CartograpHY) satellite observations over Antarctic sea ice (Schönhardt et al., 2012).

Bromine can then be recycled and re-emitted from halogen-rich condensed phases (such as sea salt aerosol or other saline solutions) or from sea ice surfaces (Pratt et al., 2013), leading to an exponential increase of bromine in the gas phase (Vogt et al., 1996). Such reactions, known as bromine explosions, lead to enhanced bromine in the atmosphere. A recent 1D chemistry model simulation predicted an increase of bromine deposition on surface snowpack after 24/48 hours of recycling over first year sea ice (Spolaor et al., 2016b). Bromine enrichment in snow (compared to sodium, relative to sea water) has therefore been recently used to reconstruct sea ice variability from ice cores both in the Antarctic and Arctic regions (Spolaor et al., 2013a, 2016b). "

Also relevant to your work is the recent atmospheric study conducted (JGR, 2016) at the coast and inland in the same East Antarctic region. In this study, both surface data and satellite observations indicate that in this region whereas the bromine chemistry is indeed maximum in spring, there is only a factor of two differences or less between spring and summer (suggesting again the importance of open ocean emissions in summer).
We believe we have addressed this topic through the previous comment.
The Legrand et al (2016) reference has been added in section 3.2.

This JGR paper also clearly showed that gaseous bromine species (that are water soluble and will be trapped in snow) largely dominate bromine aerosol. These gaseous species are likely responsible for the observed bromine enrichment in snow. Their atmospheric lifetime is far longer than the aerosol one due to a fast recycling on various surfaces (aerosol, snow grain). Therefore I have difficulty to understand the relationship between bromine enrichment in snow and the sea-ice extent ?
We point out that both gas phase and the aerosol are strongly related to sea ice. As they deposit inland, enrichment of bromine with respect to sodium would signal greater sea ice (if depleted aerosol doesn't dominate the gas phase, see a similar comment made by Referee1). Again, the model study in Spolaor 2016b shows enhancement of bromine as the air parcel travels on FYSI.

For instance, whereas you mentioned in line 64 the noise introduced by transport in using sodium to reconstruct sea-ice, you have also to mention that since the bromine enrichment is related to gaseous species, the non irreversible trapping of the bromine species in snow would strongly handicap their use as proxy of sea-ice: please comment and cite Thomas et al. (2011).
We agree with the Referee on the importance of bromine reemission from the snowpack. We point out that the maxima of bromine in Spring observed from the studies on

seasonality (this work and others, including Legrand 2016) suggest that bromine is not re-emitted significantly (if so, winter concentration would be higher, as for iodine). How much bromine is retained by spring/summer layers was the question addressed by Thomas 2011 and Dibb 2010 and they suggested snowpack reemission in Greenland, Summit. The recent paper by Legrand 2016 found that in (East) Antarctica inorganic bromine snowpack reemission is not significant, by combination of model-snowpit bromide measurements. The topic is now addressed in the introduction, including the related works.

"The stability of bromine in the snowpack was investigated at Summit, Greenland (Thomas et al. (2011), to explain the observed mixing ratios of BrO. Measurements in East Antarctica (Legrand et al., 2016) revealed that snowpack cannot account for the observed gas-phase inorganic bromine in the atmosphere. "

Sander, R., et al. (2003), Inorganic bromine in the marine boundary layer: A critical review, Atmos. Chem. Phys., 3, 1301–1336, doi:10.5194/acp-3-1301-2003.

Yang, X., R. A. Cox, N. J. Warwick, J. A. Pyle, G. D. Carver, F. M. O'Connor, and N. H. Savage (2005), Tropospheric bromine chemistry and its impacts on ozone: A model study, J. Geophys. Res., 110, D23311, doi:10.1029/2005JD006244.

Legrand, M., X. Yang, S. Preunkert, and N. Theys (2016), Year-round records of sea salt, gaseous, and particulate inorganic bromine in the atmospheric boundary layer at coastal (Dumont d'Urville) and central (Concordia) East Antarctic sites, J. Geophys. Res. Atmos., 121, doi:10.1002/ 2015JD024066.

Thomas, J. L., J. Stutz, B. Lefer, L. G. Huey, K. Toyota, J. E. Dibb, and R. von Glasow (2011), Modeling chemistry in and above snow at Summit, Greenland-Part 1: Model description and results, Atmos. Chem. Phys., 11, 4899–4914, doi:10.5194/acp-11-4899-2011.

**Line 57 :** You should cite here the atmospheric study conducted by Grilli et al. (2013) that showed a less active iodine chemistry in East Antarctica compared to the case of west Antarctica (Saiz-Lopez et al., 2007). Such a difference should enhance the motivation to examine iodine in snow throughout Antarctica (west and East).
Grilli, R., M. Legrand, A. Kukui, G. Méjean, S. Preunkert, and D. Romanini, First investigations of IO, BrO, and NO2 summer atmospheric levels at a coastal East Antarctic site using mode-locked cavity enhanced absorption spectroscopy, *Geophys. Res. Lett.*, 40, 1-6, doi:10.1002/grl.50154, 2013.
The reference has been added and the sentence has been rephrased:
"Grilli et al. (2013) have shown that ground based IO concentrations in Dumont d'Urville (Indian sector) were more than one order of magnitude lower than in the Atlantic sector (Halley station, Saiz-Lopez et al., 2007), consistent with greater sea ice in the latter."

**Line 61-67 :** As far as I know, the pioneering finding of a correlation between sea-ice and MSA in snow from Curran et al. (2003) was never clearly confirmed by more recent snow studies. Furthermore, several atmospheric studies reported no evidence of such a link at the decadal scale (Weller et al., 2011; Preunkert et al., 2007).
Weller, R., D. Wagenbach, M. Legrand, C. Elsässer, X. Tian-Kunze, and G. König-Langlo (2011), Continuous 25-years aerosol records at coastal Antarctica: Part 1. Inter-annual variability of ionic compounds and links to climate indices, Tellus, Ser. B, 63, 901–919, doi:10.1111/ j.1600-0889.2011.00542.x.

Preunkert, S., M. Legrand, B. Jourdain, C. Moulin, S. Belviso, N. Kasamatsu, M. Fukuchi, and T. Hirawake, Interannual variability of dimethylsulfide in air and seawater and its atmospheric oxidation by-products (methanesulfonate and sulfate) at Dumont d'Urville (Coastal Antarctica) (1999-2003), *J. Geophys. Res.*, 112, doi:10.1029/2006JD007585, 2007.

We agree and note that a paper currently in discussion with Climate of the Past addresses the issue of sea ice reconstructions from Law Dome (Vallelonga et al, http://www.clim-past-discuss.net/cp-2016-74/). Also, the suggested references have been added.

**Line 68-71 :** Why do you introduce a discussion on post-depositional effect here ? The existence of post-depositional effect would have no effect on decadal or centennial scales. Why "in particular Greenland" ? Please see and cite the recent work from Olivia Maselli (special issue in CP)

Maselli, O. J., Chellman, N. J., Grieman, M., Layman, L., McConnell, J. R., Pasteris, D., Rhodes, R. H., Saltzman, E., and Sigl, M.: Sea ice and pollution-modulated changes in Greenland ice core methanesulfonate and bromine, Clim. Past Discuss., doi:10.5194/cp-2016-49, accepted, 2016.

We don't completely agree with the Referee, as remobilization in the core can affect the stratigraphy in a non-unidirectional way. Therefore, the discussion on post-depositional effects in kept in the text.

"Greenland" has been removed, and the Maselli paper has been cited.

**Line 179 :** This number surprises me : 80% ? To what are related the missed (non sea-salt) sodium source that you consider to account for up to 20 % ? I don't think that the crustal source is large enough (at least for present-day climate), see and cite Weller et al. (2008).

Weller et al., Seasonal variability of crustal and marine trace elements in the aerosol at Neumayer station, Antarctica, Tellus 60B, 742-752, 2008.

We agree with the reviewer comment and therefore no crustal correction was applied to the total sodium concentration. But the reference is not added since it refers to measurements done at the Neumayer station, and not near the traverse location.

**Line 201-204**, please also cite the atmospheric study from Wagenbach et al. (1998)

Wagenbach, D., F. Ducroz, R. Mulvaney, L. Keck, A. Minikin, M. Legrand, J. S. Hall, and E. W. Wolff (1998), Sea-salt aerosol in coastal Antarctic regions, J. Geophys. Res., 103, 0,961–0,974, doi:10.1029/97JD01804.

The reference has been added.

**Section 3.2:** Please report excess bromine either as Br - 6.2 $10^{-3}$*Na (or with error calculation Br – 8 $10^{-3}$*Na in winter and use brome depletion factor in aerosol taken from Sander et al. 2003 or Legrand et al. 2016 for spring and summer).

The non-sea-salt bromine, nssBr has been calculated as [Br]-0.0062[Na] and reported in the new figures (new Fig. 4-5-6). Both $Br_{enr}$ and nssBr show very similar patterns because they are complementary calculations of non-sea salt Br (one absolute, the other a ratio). As in the answer to Referee#1 the core Talos Dome has been removed and it is now shown in the supplementary material on a depth scale.

**Line 288-292:** Not sure that this is right: at least for bromine the homogeneity is also related to the atmospheric lifetime.

We agree with the Reviewer that the lifetime of the species affects the air-to-snow flux, and that BrO is not the species deposited on the snowpack. However, we believe that the inorganic bromine lifetime in the gas-phase (predicted by models at least 5 days) to be much longer lived than the maximum distance here involved. To support our hypothesis, we observe bromine enrichment well above sea water ratio (>1) all across the traverse area. Therefore, we consider BrO concentration indicative of bromine recycling.

Note that both the BrO concentration spatial distribution from satellites and the calculated fluxes in the snow are annual averages.

**Figure 1.** (a) Schematic map of the traverse area and coring sites, marked with stars. The cores were drilled between Nov 20th 2013 and Jan 8th 2014 (early austral summer). (b) Maximum (left, August 2011) and minimum (right, January 2010) sea ice concentrations in the 130°E-170°W sector for the 2010-2013 time interval covered by the core records (NSIDC data from Meier et al., 2013). The traverse location is marked with an ellipse.

[Figure]

**Figure 3**. Distribution of bromine enrichment values within cores TD, 10, 9 (blue) and GV7, 8, 7, 6 (red). The dashed line indicates the seawater value ($Br_{enr} = 1$).

[Figure]

**Figure 4.** Variability of δ¹⁸O (upper panel), Na (middle top panel, left axis), Br (middle top panel, right axis), Brenr (middle bottom panel, left axis), nssBr (middle bottom panel, right axis), and I (bottom panel) in cores 10 (left) and 9 (right). Thick lines represent 3-month running means of the raw data (circles).

[Figure]

**Figure 5.** Variability of $\delta^{18}O$ (upper panel), Na (middle top panel, left axis), Br (middle top panel, right axis), Brenr (middle bottom panel, left axis), nssBr (middle bottom panel, right axis), and I (bottom panel) in cores GV7 (left) and 8 (right). Thick lines represent 3-month running means of the raw data (circles).

[Figure]

**Figure 6.** Variability of δ¹⁸O (upper panel), Na (middle top panel, left axis), Br (middle top panel, right axis), Brenr (middle bottom panel, left axis), nssBr (middle bottom panel, right axis), and I (bottom panel) in cores 7 (left) and 6 (right). Thick lines represent 3-month running means of the raw data (circles).

[Figure]

**Figure 7. (a)** Monthly values of sea ice area (blue) within the 130°E-170°W sector from 2010 to 2013 (±1σ, month variability) and daily average (24 hours) total downwelling spectral irradiance (red), calculated using the TUV model at 71° S, 158° E. Each irradiance calculation was set the 15th day of each month, in 2012. **(b)** Seasonality of annual bromine enrichment along the traverse: the monthly trend shows a seasonal feature with maximum in Spring. Each line refers to a core of the transect (±1σ, shaded blue area). The month averages are displayed in black. The systematic uncertainties associated to the dating are shown as verticals error bars. The magenta band represents the product distribution of normalized sea ice area and insolation, expressed in annual percentage. **(c)** Monthly sea ice area values (blue) from 2010 to 2013, with annual values of FYSI (red) and averaged bromine enrichment (black).

[Figure]

**Figure 9.** Mean annual fluxes of sodium (blue, left axes), bromine (red, right axes), iodine (green) and bromine enrichment values (black), as a function of distance from the Indian Ocean. Each dot represents a location along the traverse.

[Figure]

distance from Indian Ocean (km)

**Supplementary material**

**The following figure shows the measurements related to the Talos Dome core.**

**Figure.** Variability of δ¹⁸O (upper panel), Na (middle top panel, left axis), Br (middle top panel, right axis), Brenr (middle bottom panel, left axis), nssBr (middle bottom panel, right axis), and I (bottom panel) in the Talos Dome core on a depth scale. Thick lines represent 15 cm running means of the raw data (circles).

[Figure]

---

## Referee Report (RR1)

As already said, data on bromine and iodine in snow and ice are very welcome since they are still rather rare and are potentially interesting for a better understanding of the halogen chemistry at high southern latitudes in the past, for instance.

In the revised version, the authors considered most of my comments and suggestions. I appreciated that the estimates of excess bromine relative to sodium with respect to seawater composition are now shown. I therefore recommend it for publication. I just recommend the following clarification in the abstract.

First sentence of the abstract "Halogen chemistry in the polar regions occurs through the release of sea salt aerosols and other saline condensed phases from sea ice surfaces and organic compounds from algae colonies living within the sea ice environment." needs to be reworded since algae colonies are thought to influence iodine (not bromine) and conversely saline condensed phases are important for bromine (not iodine).

I think you can remove the last sentence of the abstract "The flux measurements are consistent with the uniform values of BrO and IO concentrations detected from satellites over the traverse area." I think that, since BrO is only a small fraction of the bromine family (most of them being water soluble and able to contribute to bromine present in snow), it is quite very dangerous to rely snow deposition of bromine and satellite BrO observations.

End of the review

---

## Author Response (AR2)

**Bromine, iodine and sodium in surface snow along the 2013 Talos Dome – GV7 traverse (Northern Victoria Land, East Antarctica)**

Niccolò Maffezzoli[1], Andrea Spolaor[2,3], Carlo Barbante[2,3], Michele Bertò[2], Massimo Frezzotti[4], Paul Vallelonga[1]

[1]Centre for Ice and Climate, Niels Bohr Institute, University of Copenhagen, Juliane Maries Vej 30, Copenhagen Ø 2100, Denmark
[2]Ca'Foscari University of Venice, Department of Environmental Science, Informatics and Statistics, Via Torino 155, 30170 Mestre, Venice, Italy
[3]Institute for the Dynamics of Environmental Processes, IDPA-CNR, Via Torino 155, 30170 Mestre, Venice, Italy
[4]ENEA, SP Anguillarese 301, 00123 Rome, Italy

*Correspondence to*: Niccolò Maffezzoli (maffe@nbi.ku.dk)

**Please find in this document:**

- The list of relevant changes
- Point-by-point responses to the reviews
- A marked-up manuscript version

**List of all relevant changes occured during discussion and revision**

- Minor changes have been addressed as in the two Referee comments.
- Fig. 9 has not been modified, as we think that it results blurred when imported in Word. The figure is shown below in its original resolution. We are happy to change the resolution if it is still considered poor resolution.

**Author comment on**

**Review of Bromine, iodine and sodium in surface snow along the 2013 Talos Dome – GV7 traverse (Northern Victoria Land, East Antarctica)**

**Referee #2**

The manuscript is much improved compared to the initial submission. I recommend publication following some minor corrections outlined here.
We would like to thank the Referee for reading the revised manuscript once again and for providing further comments to the text. We have modified the text following most of the provided comments.

Line 15 amend to: "Halogen chemistry in the polar regions occurs through the release of halogens from sea salt aerosol…"
Due to a similar comment of Michel (Referee3), the sentence has been rephrased: "Halogen chemistry in the polar regions occurs through the release of halogen elements from different sources. Bromine is primarily emitted from sea salt aerosols and other saline condensed phases associated with sea ice surfaces, while iodine is affected by the release of organic compounds from algae colonies living within the sea ice environment."

Line 16 amend to: "saline condensed phases associated with sea ice surfaces…"
Rephrased accordingly.

Line 17 and 18 amend to: "Measurements of halogen species in polar snow samples are limited to a few sites and there is some evidence that they are related to sea ice extent."
The sentence has been modified, but "although" has been left.

Line 48 amend to: "leading to formation of bromine monoxide, BrO, through reaction with ozone:"
Rephrased accordingly.

Lines 56 to 58: The Schoenhardt et al 2012 paper cited does not report tropospheric vertical columns, but total columns. Either amend the statement or find a different reference (although I am not aware that anyone has derived tropospheric columns for IO).
"Tropospheric" has been removed.

Line 221: amend to 0.0062
Rephrased accordingly.

Line 230: this is not the dominant sink of Br/source of HBr. Reaction with HCHO has the same rate constant, but HCHO will be present at much higher concentrations than HO2. Actually you don't need to include the reaction at all.
R5 has been removed.

Line 243: there is some confusion here. The statement that this parameter (bromine enrichment) is a marker of sea salt aerosol is contrary to what is written in Line 232, which talks of depletion being associated with sea salt aerosol.
We acknowledge the possible misunderstanding. Rephrased as "Such distribution of enrichment supports the theory that this parameter is, in these coastal sites, affected by sea ice signature."

Line 247 amend to: "they are also observed in winter, e.g. in core 8" – there are several examples, and core 8 is just one.
Rephrased accordingly.

Line 253 amend to: "high winter singularities or more extended peaks in cores GV7 and 8 respectively"
Rephrased accordingly.

Line 257 amend to: "by multiplying the sea ice concentration"
"Value" has been removed.

Line 273 amend to "with maximum in November"
Rephrased accordingly.

Line 274: While it's intuitive to think that the combination of sea ice and solar irradiance will generate the seasonality of bromine enrichment, the statement of line 274 "Such comparison suggests that the combined effect of sea ice and insolation drives the seasonality of bromine enrichment" is overstated when looking at Figure 7b. To look at just one example, bromine enrichment in October and November are effectively equal, while %SR.SI are considerably different. The authors need to be clear about the caveats when drawing their conclusions. A better way to present this would be "The combined effect of sea ice and insolation (Fig. 7b, magenta product distribution) shows similar features, with maximum in November, albeit with a much more pronounced springtime increase than seen in the bromine enrichment." Rephrased accordingly. I don't think that the statement about sea ice and insolation driving the seasonality of bromine enrichment can be supported by the data. One could find other factors with the same seasonality. For example, the seasonality of short-wave uv radiation alone (~300 nm) gives a November maximum (because of the effect of the ozone hole), so one could equally attribute that to driving seasonality of bromine enrichment. We don't really agree with this comment, as radiation alone doesn't give enrichment, as sea ice is needed.
Modified as: "Such comparison suggests that the combined effect of sea ice and insolation is related to the seasonality of bromine enrichment."

Line 317 amend to: "consistent with the homogeneous…"
Rephrased accordingly.

Line 333 again the authors are over-stating what can be drawn from their data. I'd suggest amending to "consistent with a sea ice influence in the area for the extra bromine deposition".
The sea ice influence in this area has been established from previous studies, cited along the text. Our findings are in line with such studies, therefore confirming them.

Lines 335 and 336: in line with my comments above I'd suggest softening the statement to "There is some evidence that the seasonality is linked to the combined effect of sea ice growth and sunlight, which trigger photochemistry above fresh sea ice"
We agree with the Referee and the sentence has been modified.

Figure 9 is poor quality (somehow low resolution) and needs to be improved before publishing.
We think that the Referee refers to the figure in the word document.
The figure resulted blurred because Word doesn't keep the PDF resolution. We therefore show below the original PDF file, being happy to increase the resolution if required.

**Author comment on**

**Review of Bromine, iodine and sodium in surface snow along the 2013 Talos Dome – GV7 traverse (Northern Victoria Land, East Antarctica)**

**Referee #3 (Michel Legrand)**

As already said, data on bromine and iodine in snow and ice are very welcome since they are still rather rare and are potentially interesting for a better understanding of the halogen chemistry at high southern latitudes in the past, for instance.
In the revised version, the authors considered most of my comments and suggestions. I appreciated that the estimates of excess bromine relative to sodium with respect to seawater composition are now shown. I therefore recommend it for publication.
We would like to thank Michel for reading the manuscript once again, and for pointing out the following two comments. We hope that more studies and past records in both the polar regions will improve the knowledge on halogens particularly important in perspective of a warming climate.

I just recommend the following clarification in the abstract.
First sentence of the abstract "Halogen chemistry in the polar regions occurs through the release of sea salt aerosols and other saline condensed phases from sea ice surfaces and organic compounds from algae colonies living within the sea ice environment." needs to be reworded since algae colonies are thought to influence iodine (not bromine) and conversely saline condensed phases are important for bromine (not iodine).
We agree with the comment. The sentence has been modified considering also Referee2 comment, as the following:
"Halogen chemistry in the polar regions occurs through the release of halogen elements from different sources. Bromine is primarily emitted from sea salt aerosols and other saline condensed phases associated with sea ice surfaces, while iodine is affected by the release of organic compounds from algae colonies living within the sea ice environment."

I think you can remove the last sentence of the abstract "The flux measurements are consistent with the uniform values of BrO and IO concentrations detected from satellites over the traverse area." I think that, since BrO is only a small fraction of the bromine family (most of them being water soluble and able to contribute to bromine present in snow), it is quite very dangerous to rely snow deposition of bromine and satellite BrO observations.
We agree with the comment, as it can be interpreted as a measurement of BrO. The sentence has been removed.

[revised manuscript text omitted]

distance from Indian Ocean (km)